# DenoiseRotator: Enhance Pruning Robustness for LLMs via Importance Concentration

**Tianteng Gu**[*]
Shanghai Jiao Tong University
995999277@sjtu.edu.cn

**Bei Liu**[*]
HKUST
beiliu@ust.hk

**Bo Xiao**
Meituan
xiaobo09@meituan.com

**Ke Zeng**
Meituan
zengke02@meituan.com

**Jiacheng Liu**
HKUST
jiachengliu@ust.hk

**Yanmin Qian**[†]
Shanghai Jiao Tong University
yanminqian@sjtu.edu.cn

## Abstract

Pruning is a widely used technique to compress large language models (LLMs) by removing unimportant weights, but it often suffers from significant performance degradation—especially under semi-structured sparsity constraints. Existing pruning methods primarily focus on estimating the importance of individual weights, which limits their ability to preserve critical capabilities of the model. In this work, we propose a new perspective: rather than merely selecting which weights to prune, we first redistribute parameter importance to make the model inherently more amenable to pruning. By minimizing the information entropy of normalized importance scores, our approach concentrates importance onto a smaller subset of weights, thereby enhancing pruning robustness. We instantiate this idea through DenoiseRotator, which applies learnable orthogonal transformations to the model's weight matrices. Our method can be seamlessly integrated with existing pruning techniques such as Magnitude, SparseGPT, and Wanda. Evaluated on LLaMA3, Qwen2.5, and Mistral models under 50% unstructured and 2:4 semi-structured sparsity, DenoiseRotator consistently improves perplexity and zero-shot accuracy. For instance, on LLaMA3-70B pruned with SparseGPT at 2:4 semi-structured sparsity, DenoiseRotator reduces the perplexity gap to the dense model by 58%, narrowing the degradation from 8.1 to 3.4 points. Codes are available at https://github.com/Axel-gu/DenoiseRotator.

## 1 Introduction

Recent advancements in large language models (LLMs) [33, 10, 1, 16] have significantly improved their performance in complex reasoning, multimodal processing, and extended context handling. However, their substantial model sizes and computational demands present practical challenges for deployment and inference efficiency. To mitigate these issues, a variety of model compression techniques have been explored, including quantization [14, 29, 3, 39, 40, 28], knowledge distillation [20], and pruning [31, 18, 13, 38, 37].

Among these, **pruning** stands out as an effective method for reducing parameter count and computational cost by eliminating less important weights. Traditional pruning methods [18] rely on importance score metrics—such as weight magnitude or output sensitivity—to rank parameters and prune those with the lowest scores. Notably, recent methods like SparseGPT [13] and Wanda

---

[*]Equal contribution

[†]Corresponding author

39th Conference on Neural Information Processing Systems (NeurIPS 2025).

[38] estimate the impact of removing individual parameters using Taylor approximations, justifying pruning parameters with minimal estimated impact and approximates the output deviation of the layer by summing the importance scores of the pruned parameters.

Despite their effectiveness, these methods operate within the fixed parameter space of pretrained models and focus solely on selecting which weights to prune. This paradigm does not modify the underlying distribution of parameter importance, which limits its flexibility and robustness—particularly under semi-structured (e.g., 2:4) sparsity constraints where pruning choices are restricted.

To overcome this limitation, we propose a fundamentally different perspective. Rather than merely selecting weights to prune, we aim to **reshape the distribution of importance score prior to pruning**. Specifically, we instantiate this idea through **DenoiseRotator**, a framework that applies learnable orthogonal transformations to reparameterize weight matrices, leveraging the computational invariance [2] of Transformer architectures. As illustrated in Figure 2, these transformations are trained to rotate the weight matrices in a way that concentrates the importance scores into a smaller subset of parameters, thereby enhancing pruning robustness.

The term **Denoise** reflects the principle that minimizing the information entropy of normalized importance scores—viewed as a discrete probability distribution—provides a theoretically grounded, differentiable, and permutation-invariant objective for importance concentration. Moreover, the norm-preserving property of orthogonal transformations ensures that the total importance of each layer remains invariant, allowing importance to be redistributed across parameters without being artificially introduced or lost. This property contributes to the overall stability of the algorithm.

Our key contributions are summarized as follows:

**Entropy-Guided Importance Concentration:** We propose enhancing pruning robustness by concentrating parameter importance into a small subset through minimizing information entropy of a discrete distribution consisting of normalized parameter importance.

**DenoiseRotator Framework:** We introduce **DenoiseRotator**, a concrete instantiation of our importance concentration idea, implemented via learnable orthogonal transformations that leverage the computational invariance [2] of Transformer [41] architectures.

**Plug-and-Play Compatibility:** DenoiseRotator decouples the importance concentration process from the actual pruning step. The learnable transformations are optimized independently prior to pruning, and can be plugged into any existing pruning pipeline.

**Extensive Empirical Evaluation:** We demonstrate the effectiveness of DenoiseRotator on a range of open-source LLMs, including Mistral (7B) [24], LLaMA3 (8B, 70B) [17], and Qwen2.5 (7B, 14B, 32B, 72B) [42]. Our method consistently and significantly improves pruning performance across unstructured and semi-structured sparsity patterns, reducing perplexity and improving accuracy compared to baseline pruning methods.

By combining entropy-guided importance concentration with orthogonal transformations, DenoiseRotator offers a novel and principled direction for robust model pruning.

## 2   Background

**Neural network pruning** is a widely used model compression technique that reduces model size and computational cost by eliminating parameters deemed less important. Existing approaches can be broadly categorized into two paradigms. The first class relies on heuristic metrics to estimate parameter importance and prunes those ranked lowest [31, 18, 13, 38, 44, 9, 26, 19, 45, 11]. The second class formulates pruning as a continuous optimization problem by relaxing the binary pruning mask into a differentiable form, allowing end-to-end training at the cost of significantly increased computational overhead [30, 12].

**Computational invariance** refers to the property of transformer architecture proposed in SliceGPT [2] that allows certain orthogonal transformations to be applied to their weight matrices without altering the model's output. Building upon this principle, QuaRot [3] and SpinQuant [29] apply orthogonal rotations to weight matrices and activations to facilitate low-bit quantization by redistributing outlier values more evenly. RotPruner [7] attempts to leverage this property for pruning and

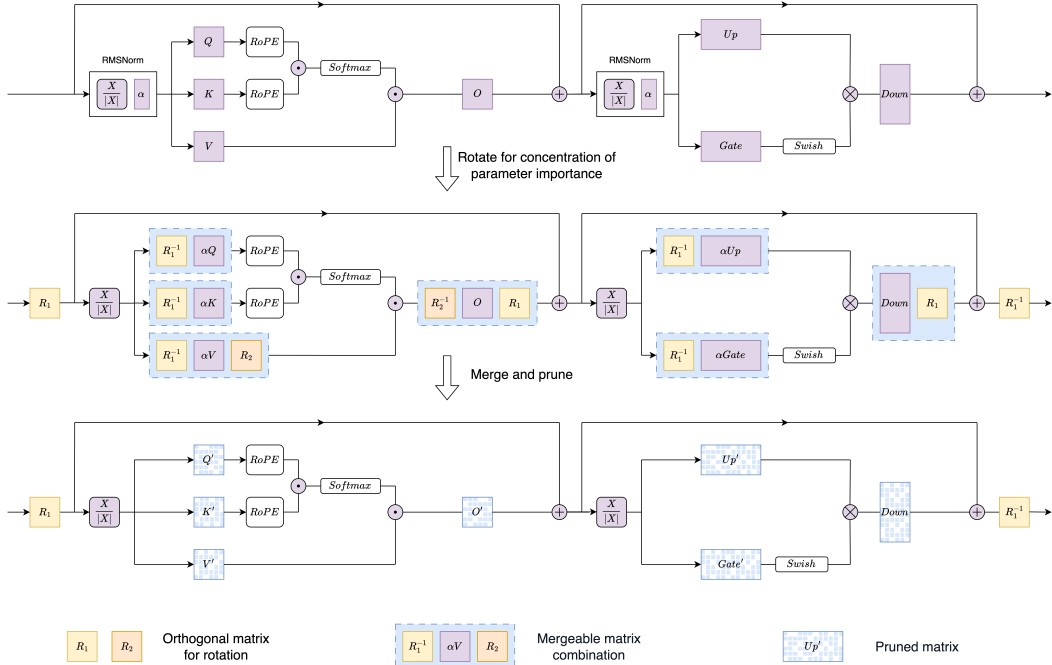

Figure 1: Overview of the DenoiseRotator framework. The top row illustrates a Transformer [41] layer architecture—used in mainstream models such as LLaMA [17], Mistral [24], and Qwen [42]—that consists of RMSNorm, attention, and feed-forward blocks. In the middle, learnable orthogonal matrices are inserted to rotate the weight matrices, concentrating parameter importance before pruning. The rotated weights are then merged and pruned in the bottom row. In this illustration, linear layers are represented in the $Y = XW$ format.

reports promising empirical results; however, it lacks theoretical analysis or formal justification for why rotation improves pruning robustness.

## 3 Method

In this section, we present the motivation and formulation of our proposed framework. We begin by formalizing the problem setting and introducing the concept of enhancing pruning robustness through concentrating parameter importance. Next, we describe how reducing the information entropy of normalized importance scores naturally leads to such concentration. Finally, we introduce a practical instantiation of this idea, termed **DenoiseRotator**, which implements entropy-based importance concentration using learnable orthogonal transformations.

### 3.1 Enhance pruning robustness via parameter importance concentration

Pruning aims to reduce model size and computational requirements by removing parameters deemed less important. A critical aspect of effective pruning is accurately estimating the importance of each parameter. As mentioned earlier, recent methods such as SparseGPT [13] and Wanda [38] leverage Taylor series to approximate the change in a linear layer output when a parameter is removed, serving as importance metrics to identify and eliminate the least significant parameters.

For instance, the Wanda method computes the importance score $S_{ij}^{\text{Wanda}}$ for each weight $W_{ij}$ of a linear layer parameterized by $W \in \mathbb{R}^{d_{\text{out}} \times d_{\text{in}}}$ by considering both the weight's magnitude and the corresponding input activation norm:

$$S_{ij}^{\text{Wanda}} = |W_{ij}| \cdot \|X_j\|_2 \tag{1}$$

This metric is equivalent to applying the Optimal Brain Damage (OBD) [26] approach to the optimization problem of pruned weight $\hat{W}$ to minimize the change in the linear layer's output after

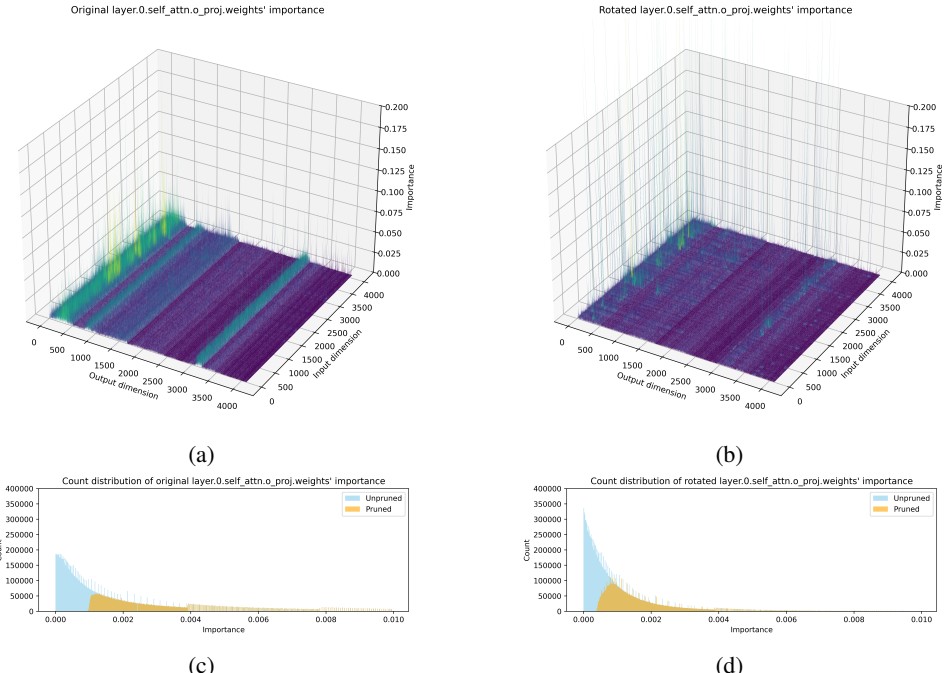

Figure 2: Visualization of OBD importance in Eq 2 for the output projection in the first layer of LLaMA-3-8B before and after orthogonal rotation. (a) and (b) show the 3D heatmaps of importance scores of the weight matrix before and after applying DenoiseRotator, respectively. (c) and (d) display the corresponding importance distributions, highlighting parameters before pruning (in blue) and after pruning (in orange) by the Wanda method. After rotation, importance becomes more concentrated.

pruning. The OBD importance score is the second-order term of Taylor approximation:

$$\min_{\hat{W}} \|WX - \hat{W}X\|^2 \quad \Rightarrow \quad S_{ij}^{\text{OBD}} = |W_{ij}|^2 \cdot (XX^\top)_{j,j} \tag{2}$$

Consequently, optimizing the change caused by pruning could be approximated by minimizing the sum of importance scores over the pruned parameters whose index is in $\mathcal{P}$, defined as the set of indices corresponding to weights selected for removal, according to Talyor approximation property:

$$\min_{\hat{W}} \|WX - \hat{W}X\|^2 \approx \min_{\mathcal{P}} \sum_{(i,j)\in\mathcal{P}} S_{ij}^{\text{OBD}} \tag{3}$$

This insight motivates minimizing the total importance of the pruned parameters, which underlies the algorithmic design of existing methods — sorting and removing the weights with the lowest importance scores. However, reducing this sum is not limited to selecting the least important parameters—we can also reshape the distribution of importance itself, offering a new perspective for enhancing pruning robustness.

To this end, we propose applying transformations to minimize total importance score of pruned weight. Formally, the optimization objective can be expressed as:

$$\min_{\hat{W},\mathcal{T}_W,\mathcal{T}_X} \left\| \mathcal{T}_W(W)\mathcal{T}_X(X) - \mathcal{T}_W(\hat{W})\mathcal{T}_X(X) \right\|^2 \approx \min_{\mathcal{P},\mathcal{T}_S} \sum_{(i,j)\in\mathcal{P}} \mathcal{T}_S(S)_{ij} \tag{4}$$

where $(\mathcal{T}_W, \mathcal{T}_X)$ is a pair of transformation applied to the layer's weight and input (e.g., orthogonal transformation), and $\mathcal{T}_S$ is the corresponding transformation applied to the importance scores $S_{ij}$. Here, we omit the specific form of the OBD score to emphasize that our method is compatible with pruning metrics based on heuristic importance estimation, where parameters with lower estimated importance are removed.

By decreasing the total importance of pruned weights, the deviation in the linear layer's output is minimized, thus preserving model performance and enhancing robustness against degradation caused by pruning. Furthermore, reducing the magnitude of importance score of pruned weights improves the accuracy of Taylor-series-based importance approximations, leading to more precise importance estimation and compensation strategies, such as those used in SparseGPT [13].

## 3.2 Reducing information entropy of normalized importance score

Although the objective in Equation 4 provides a principled formulation for importance concentration, it is generally difficult to optimize in practice. On the left-hand side, the selection of the pruning indices $\mathcal{P}$ is a combinatorial problem and is known to be NP-hard [6]. On the right-hand side, sorting-based strategies—commonly used to identify low-importance parameters—are non-differentiable and thus incompatible with gradient-based optimization.

To tackle these challenges, we propose optimizing a learnable transformation to minimize the information entropy of the normalized importance distribution prior to pruning as a surrogate objective for concentrating parameter importance. Given that importance scores are non-negative, they can be normalized to form a valid probability distribution, facilitating a probabilistic interpretation of parameter importance. Entropy, being permutation-invariant, remains unaffected by the index ordering of parameters. Furthermore, as a concave function over the probability simplex, entropy, when minimized, naturally promotes sparsity by concentrating mass onto fewer elements. This strategy independently addresses importance redistribution, separate from the pruning process itself, allowing enhanced flexibility across various pruning metrics and sparsity patterns.

Formally, let $S = \{S_{ij}\}$ denote the importance scores, and $\mathcal{T}_S$ be a learnable transformation (e.g., orthogonal rotation applied indirectly through weight and input transformation) that reshapes the distribution of importance. We first compute transformed scores $S' = \mathcal{T}_S(S)$, and normalize them over a specified group $\mathcal{G} \subseteq \{(i,j)\}$ (such as a row, column, or the entire matrix):

$$p_{ij} = \frac{\mathcal{T}_S(S)_{ij}}{\sum_{(i,j) \in \mathcal{G}} \mathcal{T}_S(S)_{ij}}, \quad \mathcal{H}(P) = -\sum_{(i,j) \in \mathcal{G}} p_{ij} \log p_{ij} \tag{5}$$

where $P = \{p_{ij}\}$ denotes the normalized importance scores computed within group $\mathcal{G}$, which could be seen as a discrete probability distribution, its entropy $\mathcal{H}(P)$ is then given by the above formula.

Our objective is to learn a shared transformation $\mathcal{T}_S$ that minimizes the entropy across all normalization groups:

$$\min_{\mathcal{T}_S} \sum_{\mathcal{G}} \mathcal{H}(P) \tag{6}$$

The summation is taken over all groups that share the same transformation $\mathcal{T}_S$. By reducing the entropy of the transformed importance distribution, we effectively transfer importance toward a smaller subset of parameters. This results in a more skewed and robust importance profile, making the model inherently more amenable to pruning.

## 3.3 DenoiseRotator

In this section, we introduce **DenoiseRotator**, a practical instantiation of the entropy-based importance concentration framework described earlier. DenoiseRotator implements the learnable transformation $\mathcal{T}_S$ using orthogonal matrices, enabling a lightweight yet effective mechanism to reshape importance distributions without altering the model's functionality.

DenoiseRotator is designed to be modular and can be easily combined with existing pruning pipelines. The following pseudocode outlines an example workflow of integrating DenoiseRotator (highlighted in blue) with layer-wise pruning methods:

---
**Algorithm 1** DenoiseRotator Integration Pipeline
---
**Require:** Dense model M, pruning method P, calibration data $D_{cal}$
**Ensure:** Pruned model M′
    Merge RMSNorm's weight into adjacent linear layers
    Compute Hessian for pruning $H \leftarrow XX^\top$ {$X$ is from $D_{cal}$}
    Integrate and train orthogonal matrices $R_1, R_2$
    Merge orthogonal matrices $R_1, R_2$
    Apply pruning method P to convert M into M′
    **return** Pruned model M′
---

### 3.3.1 Integration of orthogonal matrices

As shown in Figure 1, DenoiseRotator integrates two pairs of learnable orthogonal matrices into each Transformer layer.

**Layer-Level Rotations** ($R_1$): A pair of orthogonal matrices, $R_1$ and $R_1^\top$ of shape $(d_{hidden}, d_{hidden})$, are applied at the beginning and end of each Transformer layer, respectively. Specifically, $R_1$ is inserted before the RMS normalization and residual addition, and $R_1^\top$ is applied after these operations. $R_1$ effectively maps activations and weights into rotated spaces, allowing the network to redistribute parameter importance. Since orthogonal transformations preserve vector norms and inner products, the inputs to non-linear functions remain consistent. Thus, the model's output remains invariant.

**Attention-Level Rotations** ($R_2$): Within the self-attention, another pair of orthogonal matrices $R_2$ and $R_2^\top$ are applied to the Value ($V$) and Output ($O$) projections. These rotations adjust the internal representations, further concentrating importance and enhancing the model's resilience to pruning.

To elucidate the effect of these transformations, consider the Output ($O$) projection as an example. Let $W$ denote its weight matrix of shape $(d_{hidden}, d_{hidden})$ and $X$ its input of shape $(d_{hidden}, len \times bsz)$ before transformation. After applying the orthogonal transformations, the weight and input are transformed as follows:

$$W' = \mathcal{T}_W(W) = R_1^\top W R_2, \quad X' = \mathcal{T}_X(X) = R_2^\top X, \quad W'X' = R_1^\top W X \tag{7}$$

The corresponding transformation of the OBD importance score in Equation 2 can be expressed as:

$$S'_{ij} = \mathcal{T}_S(S)_{ij} = |W'_{ij}|^2 \cdot (X'X'^\top)_{jj} = |(R_1^\top W R_2)_{ij}|^2 \cdot \left(R_2^\top XX^\top R_2\right)_{jj} \tag{8}$$

This formulation reveals how importance scores are indirectly redistributed through orthogonal transformations applied to inputs and weights. Detailed transformation for other linear layer and pruning method could be found in Appendix A.

After training, the orthogonal matrices, except for those at the beginning and end of the Transformer layer, are merged into the model's weight matrices as illustrated in Figure 1. Pruning is then applied to the merged weight matrices.

This integration leverages structural properties of the Transformer architecture—specifically, the presence of residual connections, layer normalization (which could be transformed to RMS normalization [2]), and attention mechanisms—and is suitable for any model that exhibits similar characteristics.

### 3.3.2 Invariance of total importance

Orthogonal transformations preserve the total importance of a linear layer's weights due to their property of maintaining vector norms, i.e., $\|Rx\| = \|x\|$. Formally, this implies $\sum \mathcal{T}_S(S) = \sum S$, where $\mathcal{T}_S$ denotes the orthogonal transformation applied to the importance scores. This invariance ensures that importance is redistributed among parameters without being artificially introduced or eliminated, thereby making the algorithm stable. For a full derivation and proof of the invariance property under orthogonal transformation, see Appendix B.

To align the importance concentration mechanism with invariance, normalization in the entropy computation 5 is performed based on the position of the orthogonal matrix relative to the weight

matrix. Specifically, if the orthogonal matrix is applied on the right side (e.g., $WR$), normalization is conducted over each row. Conversely, if the orthogonal matrix is applied on the left side (e.g., $RW$), normalization is performed over each column. In cases where orthogonal matrices are applied on both sides (e.g., $R_1WR_2$), both row-wise and column-wise entropies are calculated and their sum is used. This means that the normalization group is determined by the position of the orthogonal matrix.

### 3.3.3 Optimization of the orthogonal matrix

During training, only the orthogonal matrices are optimized, while the weights of the original model and input features are kept frozen, thus the hessian matrix $H = XX^\top$ for pruning could be reused. Each orthogonal matrix is initialized as the identity matrix to ensure that the model's behavior remains unchanged at the start of training.

Since all linear layers within a Transformer layer share the same set of orthogonal transformations $R_1$ and $R_2$, the overall training objective within a Transformer layer is defined as the sum of the entropy values across all normalization groups from all affected linear layers. The total loss is:

$$\text{Loss}(R_{1,i}, R_{2,i}) = \sum_{\ell \in \mathcal{L}_i} \sum_{\mathcal{G} \in \ell} \mathcal{H}(P_\mathcal{G}) \tag{9}$$

where $\mathcal{L}_i$ denotes the set of all linear layers within the $i$-th Transformer layer, and $\mathcal{G}$ indexes the normalization groups within each linear layer. Note that we do not apply any additional weighting to the entropy of each group. This is because the number of normalization groups per linear layer is inherently tied to the weight's shape (e.g., rows or columns), and the value of discrete entropy naturally scales with the number of categories.

Maintaining the orthogonality of matrices $R_1$ and $R_2$ during backpropagation can be challenging, as direct gradient descent on these matrices does not inherently preserve orthogonality. To address this issue, we employ a reparameterization strategy using QR decomposition. For each orthogonal matrix, we define an unconstrained matrix $A$ and compute its QR decomposition: $A = QR$, where $Q$ is an orthogonal matrix and $R$ is an upper triangular matrix. During training, only the orthogonal component $Q$ is utilized in the forward pass, while gradients are applied to the underlying matrix $A$. The theoretical justification for the feasibility of this method is provided in Appendix C.

## 4  Experiment

**Models and Tasks**   We evaluate our method on several recent open-source large language models, including Mistral 7B [24], LLaMA 3 (8B and 70B) [17], and Qwen2.5 (7B, 14B, 32B, and 72B) [42]. Our evaluation encompasses both language generation, measured by perplexity on the WikiText-2 dataset [32], and five widely-used zero-shot tasks: PIQA [5], WinoGrande [36], HellaSwag [43], ARC-e, and ARC-c [8]. Following established practices, we utilize the LM Evaluation Harness [15] with default settings for all evaluations.

**Baselines**   We integrate DenoiseRotator with three pruning methods: the classic Magnitude pruning and two advanced techniques, Wanda and SparseGPT. Both Wanda and SparseGPT require calibration data to estimate input statistics. For consistency, we use the same calibration dataset as in prior work, consisting of 128 sequences with a context length of 2048 tokens, sampled from the C4 training set [35]. We apply a uniform sparsity level across all decoder layers and evaluate two types of sparsity: unstructured 50% sparsity and semi-structured 2:4 sparsity.

**Setup**   We trained the orthogonal matrices of DenoiseRotator using the Adam optimizer with a learning rate of 0.01 over 2000 steps. All computations, except for QR decomposition, were performed in torch.bfloat16 precision to enhance efficiency. The computation resource cost scales linearly with the number of model parameters. For instance, training on LLaMA 3 70B with SparseGPT took approximately 28 hours and utilized around 30 GB of GPU memory on a single NVIDIA A100 GPU. Notably, the training duration is independent of the calibration dataset size, as the optimization process reuse the hessian matrix for pruning. No recovery finetuning was performed after pruning.

## 4.1 Main result

In Table 1, we evaluate the generation capabilities of both dense and pruned models—with and without the integration of DenoiseRotator—using perplexity on the WikiText-2 validation set. Under both unstructured (50%) and semi-structured (2:4) sparsity settings, DenoiseRotator consistently and significantly reduces perplexity across all model families and scales.

Table 1: Perplexity ↓ results on WikiText-2 of various models under different sparsity settings.

| Sparsity | Method | Mistral | LLaMA3 | | Qwen2.5 | | | |
| | | 7B | 8B | 70B | 7B | 14B | 32B | 72B |
|---|---|---|---|---|---|---|---|---|
| 0% | Dense | 5.95 | 6.14 | 2.86 | 6.85 | 5.29 | 5.02 | 3.88 |
| 50% | Magnitude | 30.39 | 30.39 | 10.58 | 198.88 | 22.94 | 19.22 | 734.04 |
| | +DenoiseRotator | 7.30 | 14.43 | 7.00 | 9.27 | 8.78 | 6.97 | 5.37 |
| | Wanda | 6.92 | 9.86 | 5.80 | 8.61 | 7.31 | 6.30 | 5.22 |
| | +DenoiseRotator | 6.52 | 7.82 | 4.73 | 7.93 | 6.72 | 5.99 | 4.94 |
| | SparseGPT | 6.94 | 9.57 | 5.99 | 8.46 | 7.27 | 6.35 | 4.94 |
| | +DenoiseRotator | **6.38** | **7.60** | **4.61** | **7.60** | **6.51** | **5.86** | **4.78** |
| 2:4 | Magnitude | 141.96 | 141.96 | 18.17 | 559.87 | 58.93 | 24.27 | 287.70 |
| | +DenoiseRotator | 9.52 | 75.23 | 11.53 | 11.97 | 13.51 | 8.61 | 8.81 |
| | Wanda | 10.18 | 25.19 | 9.39 | 15.01 | 11.66 | 8.08 | 6.69 |
| | +DenoiseRotator | 7.80 | 11.41 | 6.60 | 10.13 | 8.71 | 7.90 | 6.16 |
| | SparseGPT | 9.71 | 17.67 | 10.97 | 11.35 | 10.20 | 7.92 | 7.19 |
| | +DenoiseRotator | **7.30** | **10.01** | **6.25** | **8.88** | **7.86** | **6.75** | **5.85** |

We report zero-shot task results for dense and pruned models in Table 2. Across both unstructured (50%) and semi-structured (2:4) sparsity settings, DenoiseRotator consistently improves zero-shot accuracy, often recovering performance close to or surpassing the dense model, especially on larger models like Qwen2.5-72B and LLaMA3-70B.

Table 2: Average zero-shot accuracy (%) ↑ on five benchmark tasks (PIQA, WinoGrande, HellaSwag, ARC-e, ARC-c).

| Sparsity | Method | Mistral | LLaMA3 | | Qwen2.5 | | | |
| | | 7B | 8B | 70B | 7B | 14B | 32B | 72B |
|---|---|---|---|---|---|---|---|---|
| 0% | Dense | 74.21 | 72.72 | 80.05 | 72.18 | 75.81 | 75.12 | 78.69 |
| 50% | Magnitude | 57.78 | 57.78 | 64.05 | 39.09 | 57.52 | 63.81 | 41.27 |
| | +DenoiseRotator | 69.79 | 60.44 | 70.37 | 66.83 | 62.45 | 69.19 | 75.54 |
| | Wanda | 71.21 | 65.77 | 76.23 | 67.28 | 73.36 | 73.82 | 78.04 |
| | +DenoiseRotator | 72.66 | 69.28 | 78.37 | 71.35 | 75.18 | **76.39** | 78.32 |
| | SparseGPT | 71.76 | 66.88 | 76.66 | 68.95 | 74.00 | 74.06 | 78.25 |
| | +DenoiseRotator | **73.46** | **69.58** | **78.54** | **72.18** | **75.71** | 76.02 | **78.42** |
| 2:4 | Magnitude | 48.04 | 48.04 | 59.04 | 40.51 | 53.89 | 59.79 | 41.41 |
| | +DenoiseRotator | 60.57 | 41.10 | 61.94 | 62.41 | 58.89 | 66.45 | 69.42 |
| | Wanda | 63.28 | 51.03 | 70.09 | 61.24 | 64.66 | 71.09 | 74.94 |
| | +DenoiseRotator | 69.61 | 64.22 | 75.74 | 67.85 | 70.23 | **73.37** | **77.66** |
| | SparseGPT | 66.18 | 55.95 | 69.16 | 64.67 | 66.76 | 71.10 | 75.43 |
| | +DenoiseRotator | **70.98** | **66.11** | **76.37** | **69.56** | **72.38** | 73.25 | 77.16 |

Although we do not explicitly optimize for the 2:4 semi-structured sparsity constraint, DenoiseRotator still performs remarkably well—even making previously unusable models viable under this constraint. This is likely because the orthogonal matrices act as a form of random permutation, increasing the chance that crucial weights align with the semi-structured sparsity pattern.

The impact of rotation is illustrated in Figure 2, which visualizes the OBD importance for the output projection in the first layer of LLaMA-3-8B before and after applying DenoiseRotator, revealing how importance becomes more concentrated after transformation.

For detailed performance results (including the LLaMA 1 and 2 families), please refer to Appendix G.

## 4.2 Effectiveness of Entropy Reduction on Pruning Robustness

To validate the efficacy of entropy reduction in enhancing pruning robustness, we conduct an ablation study on LLaMA-3-8B using SparseGPT with a 50% unstructured sparsity calibrated by 128 sequences of 2048 tokens sampled from C4 training set (generation task). We record model performance at various entropy reduction steps of DenoiseRotator. Table 3 presents the average entropy value per layer, time

Table 3: Impact of entropy reduction on pruning robustness. Results are reported on LLaMA-3-8B with SparseGPT (unstructured 50% sparsity) calibrated by 128 sequences of 2048 tokens sampled from C4 training set.

| Step | Entropy | Time (s) | Zero-shot (%) ↑ | Wikitext2 PPL ↓ |
|------|---------|----------|-----------------|-----------------|
| 0    | 457280  | 1181     | 66.88           | 9.567           |
| 100  | 396992  | 1603     | 70.54           | 7.701           |
| 400  | 387904  | 2760     | 70.12           | 7.619           |
| 2000 | 384128  | 9120     | 69.58           | 7.597           |

consumed, average zero-shot accuracy, and perplexity on the WikiText2 validation set over different optimization steps. 0 step refers to the SparseGPT baseline.

As shown in Table 3, reducing the entropy of the normalized importance distribution significantly improves model's performance. At step 100, where average entropy per layer is reduced by 13%, we observe a substantial boost in zero-shot accuracy from 66.88% to 70.54%, along with a decrease in perplexity from 9.567 to 7.701. Further optimization continues to reduce entropy and improve perplexity. A comprehensive hyperparameter analysis, detailing optimization steps and learning rates, is available in Appendix D.

These results empirically support our hypothesis: concentrating importance via entropy reduction enables more robust pruning, leading to reduced performance degradation.

## 4.3 Inference speedup and overhead analysis

DenoiseRotator introduces a pair of orthogonal matrices at the beginning and end of each layer. Since the orthogonal matrices of adjacent layers can be merged, the actual overhead is limited to storing an additional shape matrix (hidden_size, hidden_size) per layer, along with a matrix multiplication during inference. For example, in LLaMA-3-8B, this results in approximately 0.5 billion additional parameters (i.e., $4096 \times 4096 \times 32$). This overhead is relatively small, accounting for only about 6.7% of the original model size. To further address overhead concerns, we explore the potential of block diagonal orthogonal matrices as a reduction strategy in Appendix E.

Table 4: Average inference time and speedup per layer on LLaMA3-8B for 32 sequences of length 2048 on A100 GPU

| Configuration | Dense Layer | 2:4 Sparse Layer | + Orthogonal Matrix |
|---------------|-------------|------------------|---------------------|
| **Time (ms)** | 5.80 | 4.37 | 4.69 |
| **Speedup (%)** | 1.00× (baseline) | 1.33× | 1.24× |

We evaluate the average inference time of a single Transformer layer in LLaMA3-8B on 32 sequences of length 2048 using an NVIDIA A100 GPU. The batch size is set to 1, and we utilize PyTorch's semi-structured sparsity framework, specifically the `torch.sparse.to_sparse_semi_structured` function. Timing measurements are conducted using `torch.utils.benchmark.Timer`. As shown in Table 4, a dense layer takes 5.80 ms on average, while a 2:4 semi-structured sparse layer reduces this to 4.37 ms—achieving a 1.33× speedup. The additional cost introduced by DenoiseRotator's orthogonal matrix is only 0.32 ms per layer, which is minor compared to the overall computation time and maintains practical efficiency for deployment.

## 5   Conclusion

This paper introduces DenoiseRotator, a plug-and-play framework that improves the robustness of model pruning through importance concentration. By applying learnable orthogonal transformations optimized via entropy reduction, DenoiseRotator reshapes the importance landscape to ensure critical informations and abilities are preserved during pruning. Our theoretical analysis and empirical studies validate the effectiveness of this strategy: concentrating importance distributions results in more informative retained parameters and enhances the model's robustness to pruning. Extensive evaluations on multiple LLMs show consistent gains across perplexity and zero-shot accuracy under both unstructured and semi-structured sparsity. Moreover, DenoiseRotator achieves these improvements with minimal overhead and without modifying the original pruning pipeline. These results demonstrate the promise of entropy-guided importance reshaping as a general principle for robust and efficient model sparsification. While our experiments focus on post-training pruning, the underlying idea of concentrating parameter importance may also benefit other stages of the model lifecycle, such as being integrated into pretraining or continual training.

## 6   Limitations

Semi-structured sparsity patterns offer hardware acceleration benefits but also impose structural constraints. Future work is needed to explore how such constraints can be explicitly integrated into the learnable transformation's optimization process.

# 7 Acknowledgements

This work was supported by the China STI 2030-Major Projects under Grant No. 2021ZD0201500, and it was also supported in part by the Beijing Nova Program.

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

# A  Details of Orthogonal Transformations and Importance Score Formulations

In this section, we present how orthogonal transformations affect the importance scores under three commonly used pruning methods: Magnitude pruning, Wanda, and SparseGPT. We also show how we modify the importance formulas to maintain the Taylor expansion property under transformation.

Throughout this discussion, we denote the original weight matrix as $W \in \mathbb{R}^{d_{\text{out}} \times d_{\text{in}}}$, the input matrix as $X \in \mathbb{R}^{d_{\text{in}} \times n}$, and approximate the Hessian as $H = XX^\top$. The output of a linear layer is given by $Y = WX$.

Depending on which sides the orthogonal transformations are applied, three typical cases arise:

**Case 1: Right-side rotated weight and left side rotated input (e.g., Q, K, Up, Gate).**

For layers such as Query (Q), Key (K), Up, and Gate, the weight is rotated on the right side and the input on the left. Specifically, the transformed weights and inputs are given by $W' = WR_1$ and $X' = R_1^\top X$, respectively. The corresponding changes in importance scores under different pruning methods are summarized below:

Table 5: Right-side rotated weight and left-side rotated input (e.g., Q, K, Up, Gate)

| Method | Original Importance Score $S_{ij}$ | Transformed Importance Score $\mathcal{T}_S(S_{ij})$ |
|---|---|---|
| Magnitude | $\lvert W_{ij} \rvert$ | $(WR_1)_{ij}^2$ |
| Wanda | $\lvert W_{ij} \rvert \cdot \lVert X_j \rVert_2$ | $(WR_1)_{ij}^2 \cdot (R_1^\top H R_1)_{jj}$ |
| SparseGPT | $W_{ij}^2 \,/\, H_{jj}^{-1}$ | $(WR_1)_{ij}^2 \,/\, (R_1^\top H^{-1} R_1)_{jj}$ |

**Case 2: Left-side rotated weight and unrotated input (e.g., Down)**

For layers like Down, only a left-side rotation is applied to the weights, while the inputs remain unrotated. In this case, $W' = R_1^\top W$ and $X' = X$. The modified importance scores are given by:

Table 6: Left-side rotated weights and unrotated inputs (e.g., Down)

| Method | Original Importance Score $S_{ij}$ | Transformed Importance Score $\mathcal{T}_S(S_{ij})$ |
|---|---|---|
| Magnitude | $\lvert W_{ij} \rvert$ | $(R_1^\top W)_{ij}^2$ |
| Wanda | $\lvert W_{ij} \rvert \cdot \lVert X_j \rVert_2$ | $(R_1^\top W)_{ij}^2 \cdot H_{jj}$ |
| SparseGPT | $W_{ij}^2 \,/\, H_{jj}^{-1}$ | $(R_1^\top W)_{ij}^2 \,/\, H_{jj}^{-1}$ |

**Case 3: Two-side rotated weight and left-side rotated input (e.g., O, V)**

For components like Output (O) and Value (V) projections in the self-attention mechanism, rotations are applied on both sides.

For Output (O): $W' = R_2^\top W R_1$ and $X' = R_1^\top X$. The transformed importance scores are:

Table 7: Transformation for Output (O)

| Method | Original Importance Score $S_{ij}$ | Transformed Importance Score $\mathcal{T}_S(S_{ij})$ |
|---|---|---|
| Magnitude | $\lvert W_{ij} \rvert$ | $(R_2^\top W R_1)_{ij}^2$ |
| Wanda | $\lvert W_{ij} \rvert \cdot \lVert X_j \rVert_2$ | $(R_2^\top W R_1)_{ij}^2 \cdot (R_1^\top H R_1)_{jj}$ |
| SparseGPT | $W_{ij}^2 \,/\, H_{jj}^{-1}$ | $(R_2^\top W R_1)_{ij}^2 \,/\, (R_1^\top H^{-1} R_1)_{jj}$ |

For Value (V): $W' = R_1^\top W R_2$ and $X' = R_2^\top X$. The corresponding transformation is similar to that of Output (O).

# B  Proof of invariance of total importance

As established in Equation 3, the total importance of parameters can be approximated by the Frobenius norm of the layer output, corresponding to the scenario where all weights are pruned (set to zero). Specifically, according to optimal brain damage [26], we have:

$$\sum_{i,j} \left( S_{ij} + \mathcal{O}(|W_{ij}|^3) \right) = \|WX\|^2 \quad \Rightarrow \quad \sum S \approx \|WX\|^2 \tag{10}$$

Since the magnitudes of $|W_{ij}|$ are very small, this approximation is highly accurate.

**Case 1: Right side rotated weights and left side rotated inputs (e.g., Q, K, Up, Gate).**

When both the weight matrix and the input are transformed via orthogonal matrices:

$$\Sigma T_S(S) \approx \|\mathcal{T}_W(W)\mathcal{T}_X(X)\| = \|W R_1 R_1^\top X\| = \|WX\| \approx \Sigma S \tag{11}$$

**Case 2: Left side rotated Weights and unrotated input (e.g., Down).**

When only the weight is rotated:

$$\Sigma T_S(S) \approx \|\mathcal{T}_W(W)\mathcal{T}_X(X)\| = \|R_1^\top WX\| = \mathrm{tr}(W^\top X^\top R_1 R_1^\top WX) = \|WX\| \approx \Sigma S \tag{12}$$

This equality holds because orthogonal matrices satisfy $R^\top = R^{-1}$, and hence preserve both vector norms and inner products.

**Case 3: Two side rotated weight and left side rotated input (e.g., O, V).**

Components such as $O$ and $V$, which are affected by both left and right orthogonal transformations, can be viewed as a combination of the two cases above. Thus, their total importance is also preserved.

# C  QR decomposition reparameterization

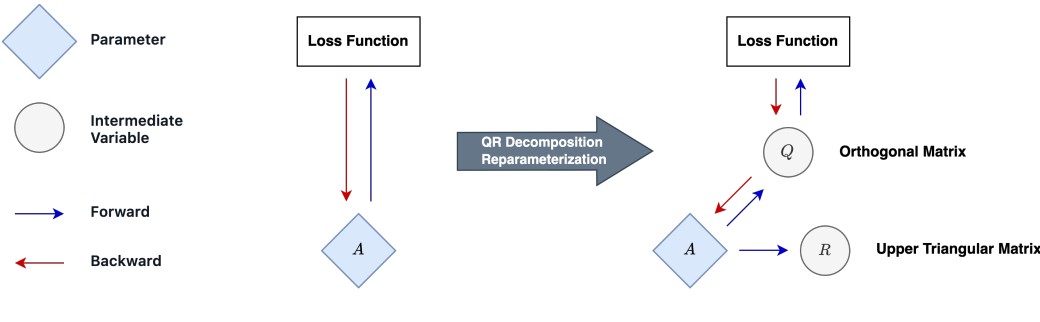

Figure 3: Illustration of the QR decomposition reparameterization process. The diagram shows how an unconstrained matrix $A$ is optimized indirectly to achieve constrained optimization of the orthogonal matrix $Q$, while interfacing with the loss function during forward and backward passes. The process leverages QR decomposition to preserve orthogonality and integrate seamlessly into gradient-based optimization methods.

Standard gradient descent algorithms and their variants encounter significant challenges when dealing with the orthogonality constraint of matrices. In conventional settings, these algorithms directly update matrix parameters without ensuring that the orthogonality condition is preserved post-update. This can lead to a loss of orthogonality, compromising model integrity and stability. To address this, previous research [29, 22, 7] has frequently adopted Stiefel manifold optimization techniques [23, 27, 4, 25]. These methods are specifically tailored to handle the complex geometric constraints associated with orthogonal matrices, necessitating specialized optimizers and additional computational resources.

QR decomposition provides an elegant solution to this problem by exploiting the properties of linear algebra to indirectly parameterize orthogonal matrices. This approach sidesteps the orthogonality constraint by using the QR decomposition, thus enabling the use of conventional gradient descent methods for optimization tasks without requiring complex manifold optimizers. As a result, QR decomposition not only maintains the orthogonality of matrices but also simplifies the optimization process, effectively mitigating the difficulties posed by orthogonality constraints.

In Figure 3, we start with an unconstrained matrix $A$. During the forward pass of the neural network, a QR decomposition is performed, splitting $A$ into two matrices: an orthogonal matrix $Q$ and an upper triangular matrix $R$. The orthogonal matrix $Q$ becomes the immediate subject of optimization in the loss function, even though we never directly modify $Q$ itself. Instead, we focus on optimizing the unconstrained matrix $A$, which indirectly alters $Q$ due to the nature of QR decomposition.

The process of QR decomposition is composed of basic matrix operations like dot products, vector normalization, and projection. These are mathematically smooth operations, meaning they can be differentiated, which is key for gradient-based optimization. Each step in QR decomposition involves operations that smoothly vary, making it possible to calculate how small changes in the input $A$ affect the output $Q$ and $R$.

Because of these properties, QR decomposition can seamlessly integrate into the computation graph used by automatic differentiation tools. PyTorch [34] provides automatic differentiation support for QR factorization through `torch.qr`. When QR decomposition is performed, PyTorch automatically manages and propagates gradients with respect to the input matrix $A$. This capability enables gradient-based methods, such as gradient descent, to efficiently update $A$, automatically respecting the orthogonality constraint imposed on $Q$.

QR decomposition reparameterization therefore transcends the limitations of direct parameterization by introducing a flexible and effective mechanism to handle orthogonality in optimization tasks. As the network trains, the orthogonal matrix $Q$ adapts continuously to minimize the loss function, while the underlying matrix $A$ undergoes refinement through regular gradient-based updates.

# D   Hyperparameter Sensitivity Analysis

A thorough analysis of hyperparameter sensitivity is essential for understanding how different hyperparameter choices impact the final performance of a model. In the main paper, we relied on a fixed set of hyperparameters, such as using 2000 steps in the rotation training process. To enhance the rigor and reproducibility of our research, we conducted comprehensive experiments on the LLaMA-3-8B model, exploring the effects of various hyperparameter configurations on the final results.

In our experiments, each row presents the perplexity and average zero-shot accuracy corresponding to different learning rates and training steps. The setup involves utilizing SparseGPT with 50% unstructured pruning, using a calibration dataset consisting of 128 samples of length 2048 from the C4 dataset. All experiments were performed on an A100 80G GPU.

Table 8: Results across various learning rates and training steps

| Steps | 0.1 | | 0.01 | | 0.001 | | 0.0001 | |
|---|---|---|---|---|---|---|---|---|
| | Perplexity | Zero-shot | Perplexity | Zero-shot | Perplexity | Zero-shot | Perplexity | Zero-shot |
| 100 | 7.75 | 69.52 | 7.70 | 70.54 | 7.96 | 69.95 | 8.52 | 67.83 |
| 200 | 7.70 | 70.31 | 7.64 | 68.93 | 7.82 | 70.04 | 8.30 | 68.59 |
| 400 | 7.63 | 69.84 | 7.62 | 70.12 | 7.66 | 70.31 | 8.16 | 69.37 |
| 800 | 7.61 | 70.25 | 7.61 | 70.21 | 7.72 | 69.55 | 7.94 | 70.77 |
| 2000 | 7.64 | 69.82 | 7.60 | 69.58 | 7.67 | 70.36 | 7.85 | 70.11 |
| 4000 | 7.60 | 70.46 | 7.60 | 69.59 | 7.63 | 70.37 | 7.78 | 69.59 |

These results indicate: **1**. When the learning rate is less than or equal to 0.001, the efficiency is low due to the small learning rate. **2**. With a learning rate of 0.01, training converges around 2000 steps, and excessive steps increase training costs. **3**. Compared to the baseline without the importance concentration mechanism, all configurations show significant improvements as training progresses. **4**. Zero-shot task accuracy fluctuates around 70%, with a variation of approximately 1%, not showing a clear linear relationship with perplexity.

# E  Trade-off with Block Diagonal Orthogonal Matrices

Block diagonal orthogonal matrices provide an effective approach to reduce both training and inference overhead of DenoiseRotator. By leveraging the structural efficiency of block diagonal matrices, this method achieves a balanced optimization of performance and resource utilization.

In our experiments with the LLaMA-3-8B model, we evaluated these benefits using SparseGPT with 50% unstructured pruning over 2000 steps and a learning rate of 0.01. Dense orthogonal matrices were replaced with block diagonal matrices, consisting of block_num orthogonal submatrices arranged along the diagonal. Each submatrix has dimensions of $\frac{\text{hidden\_size}}{\text{block\_num}}$. This approach reduces both spatial and computational complexity to $\frac{1}{\text{block\_num}}$ of the original requirements.

Table 9: Performance and Computational Costs with Different Block Configurations

| Block Number | Perplexity | Zero-Shot Accuracy | Time Cost per Step (s) | Entropy |
|:---:|:---:|:---:|:---:|:---:|
| 1 | 7.597 | 69.58 | 0.124 | 384128 |
| 2 | 8.024 | 68.68 | 0.088 | 410816 |
| 4 | 8.544 | 68.47 | 0.076 | 428160 |
| 8 | 8.882 | 67.51 | 0.072 | 440512 |

These results highlight the trade-offs between performance and computational demands across different block configurations, illustrating the capacity of block diagonal orthogonal matrices to optimize these aspects effectively.

Looking ahead, potential enhancements could involve: **1**. Varying the dimensions of each block within the block diagonal matrix (as opposed to maintaining uniform block sizes in this section's experiments) to further improve performance. **2**. Incorporating permutation matrices, which have low computational complexity, to combine with block diagonal orthogonal matrices, thus emulating dense orthogonal matrices. This approach might involve arranging highly correlated dimensions or parameters adjacently for enhanced efficacy.

# F  Compatibility with LoRA for Fine-Tuning Pruned Models

In this section, we explore the use of LoRA [21] to fine-tune pruned models. By freezing rotation matrices during fine-tuning, we maintain consistent gradient flow, aligning with scenarios where rotations are not applied, thus ensuring stability throughout the training process.

**DenoiseRotator Configuration**

- Steps: 100
- Learning Rate: 0.01
- Model: LLaMA-3-8B
- Pruning Method: Wanda, 2:4 semi-structured

**Fine-tuning Configuration**

- Method: LoRA
- Alpha: 32.0
- Dropout: 0.1
- LoRA-r: 8
- Dataset: 4096 samples of length 2048 from the WikiText2 train set
- Learning Rate: 2e-4
- Weight Decay: 1e-2
- Optimizer: Adam
- Learning Rate Scheduler: Linear

- Warm-up Steps: 400
- Note: Rotation matrices are kept frozen during fine-tuning

Table 10: Performance of Fine-Tuning with LoRA on Pruned Models

| Method | Perplexity | Zero-Shot Accuracy |
|---|---|---|
| Dense | 6.14 | 72.72 |
| Wanda | 25.19 | 51.03 |
| + DenoiseRotator | 14.31 | 60.67 |
| + Finetune | 9.32 | 61.08 |

These results illustrate that additional fine-tuning with LoRA further reduces perplexity and improves accuracy. This suggests that fine-tuning post-denoising is an effective strategy for boosting performance, while confirming that the rotation matrices introduced do not destabilize the training process.

# G   Performance of pruned model

Table 11: Pruning LLaMA - 3.2 - 1B

| Model | | | | | | | Perplexity | |
|---|---|---|---|---|---|---|---|---|
| LLaMA - 3.2 - 1B | ARC-c | ARC-e | Hellaswag | Piqa | Winogrande | Average | wikitext2 | c4 |
| **0%** Dense | 36.77 | 60.52 | 63.65 | 74.43 | 60.3 | 59.14 | 9.748 | 14.034 |
| **50%** Magnitude | 22.35 | 28.87 | 30.51 | 55.6 | 48.38 | 37.14 | 410.099 | 535.319 |
| +DenoiseRotator | 24.32 | 34.68 | 35.4 | 58.0 | 52.41 | 40.96 | 55.735 | 79.761 |
| Wanda | 26.02 | 47.77 | 44.1 | 64.64 | 54.85 | 47.48 | 23.881 | 35.242 |
| +DenoiseRotator | 29.78 | 49.62 | 54.28 | 70.67 | 56.35 | 52.14 | 13.98 | 20.213 |
| SparseGPT | 29.61 | 47.52 | 50.63 | 68.06 | 57.06 | 50.58 | 19.961 | 28.117 |
| +DenoiseRotator | **33.02** | **52.48** | **57.01** | **71.33** | **59.04** | **54.58** | **12.857** | **18.611** |
| **2:4** Magnitude | 24.57 | 26.52 | 26.92 | 52.39 | 49.49 | 35.98 | 7417.287 | 6169.094 |
| +DenoiseRotator | 22.18 | 31.23 | 29.75 | 55.77 | 51.07 | 37.99 | 458.362 | 524.964 |
| Wanda | 22.53 | 36.32 | 31.53 | 57.62 | 50.36 | 39.67 | 90.658 | 131.275 |
| +DenoiseRotator | 25.34 | 46.25 | 42.72 | 65.45 | 52.33 | 46.42 | 25.896 | 36.962 |
| SparseGPT | 25.0 | 40.57 | 38.06 | 61.32 | 53.91 | 43.77 | 36.233 | 48.101 |
| +DenoiseRotator | **28.67** | **48.74** | **48.62** | **68.28** | **57.46** | **50.35** | **18.088** | **25.091** |
| **4:8** Magnitude | 23.89 | 28.24 | 27.77 | 52.88 | 47.99 | 36.18 | 1257.331 | 1334.822 |
| +DenoiseRotator | 23.29 | 29.55 | 30.59 | 55.11 | 52.17 | 38.14 | 245.479 | 259.474 |
| Wanda | 22.87 | 39.77 | 35.59 | 62.35 | 49.41 | 42.0 | 48.225 | 69.767 |
| +DenoiseRotator | 29.01 | 50.34 | 48.68 | 68.39 | 55.09 | 50.3 | 18.199 | 26.491 |
| SparseGPT | 25.51 | 45.33 | 43.37 | 65.18 | 56.04 | 47.09 | 25.458 | 35.121 |
| +DenoiseRotator | **31.14** | **50.63** | **52.49** | **70.29** | **56.27** | **52.17** | **15.019** | **21.369** |

Table 12: Pruning LLaMA - 3 - 8B

| Model | Zero-shot accuracy (%) | | | | | | Perplexity | |
|---|---|---|---|---|---|---|---|---|
| LLaMA - 3 - 8B | ARC-c | ARC-e | Hellaswag | Piqa | Winogrande | Average | wikitext2 | c4 |
| 0% Dense | 52.73 | 77.78 | 79.19 | 80.96 | 72.93 | 72.72 | 6.138 | 9.443 |
| 50% Magnitude | 35.92 | 58.46 | 59.33 | 72.2 | 62.98 | 57.78 | 30.393 | 36.197 |
| +DenoiseRotator | 38.65 | 60.02 | 65.51 | 73.12 | 64.88 | 60.44 | 14.425 | 22.761 |
| Wanda | 44.88 | 67.97 | 68.8 | 76.71 | 70.48 | 65.77 | 9.864 | 15.243 |
| +DenoiseRotator | **48.12** | 72.73 | 74.82 | 78.89 | 71.82 | 69.28 | 7.816 | 12.251 |
| SparseGPT | 44.45 | 69.44 | 72.34 | 76.88 | 71.27 | 66.88 | 9.567 | 14.223 |
| +DenoiseRotator | 46.67 | **74.45** | **75.67** | **78.94** | **72.14** | **69.58** | **7.597** | **11.819** |
| 2:4 Magnitude | 30.03 | 42.68 | 46.25 | 66.38 | 54.85 | 48.04 | 141.962 | 183.124 |
| +DenoiseRotator | 23.38 | 36.45 | 37.11 | 57.51 | 51.07 | 41.1 | 75.233 | 107.187 |
| Wanda | 30.38 | 50.63 | 47.78 | 67.9 | 58.48 | 51.03 | 25.189 | 37.082 |
| +DenoiseRotator | 41.64 | **70.16** | 66.38 | 74.1 | 68.82 | 64.22 | 11.411 | 17.657 |
| SparseGPT | 32.76 | 55.72 | 55.97 | 70.08 | 65.19 | 55.95 | 17.669 | 25.027 |
| +DenoiseRotator | **41.89** | 70.12 | **69.67** | **76.88** | **71.98** | **66.11** | **10.008** | **15.011** |
| 4:8 Magnitude | 32.42 | 49.92 | 53.67 | 70.08 | 59.91 | 53.2 | 47.863 | 59.894 |
| +DenoiseRotator | 27.73 | 44.78 | 53.92 | 66.43 | 59.67 | 50.51 | 23.971 | 34.597 |
| Wanda | 36.69 | 59.13 | 59.28 | 71.76 | 66.14 | 58.6 | 14.453 | 21.78 |
| +DenoiseRotator | **44.97** | **72.69** | 71.09 | 76.66 | 71.19 | 67.32 | 9.274 | 14.569 |
| SparseGPT | 37.12 | 61.53 | 63.87 | 74.32 | 67.56 | 60.88 | 12.836 | 18.443 |
| +DenoiseRotator | 44.62 | 68.9 | **73.14** | **77.64** | **73.48** | **67.56** | **8.557** | **13.151** |

Table 13: Pruning LLaMA-3-70B

| Model | Zero-shot accuracy (%) | | | | | | Perplexity | |
|---|---|---|---|---|---|---|---|---|
| LLaMA-3-70B | ARC-c | ARC-e | Hellaswag | Piqa | Winogrande | Average | wikitext2 | c4 |
| 0% Dense | 64.25 | 85.9 | 84.87 | 84.55 | 80.66 | 80.05 | 2.857 | 7.167 |
| 50% Magnitude | 43.6 | 68.06 | 70.4 | 73.78 | 64.4 | 64.05 | 10.578 | 17.681 |
| +DenoiseRotator | 48.38 | 73.02 | 76.9 | 78.73 | 74.82 | 70.37 | 6.998 | 11.951 |
| Wanda | 58.87 | 81.48 | 80.97 | 82.64 | 77.19 | 76.23 | 5.798 | 9.861 |
| +DenoiseRotator | 60.58 | 84.39 | 83.24 | 83.62 | 80.03 | 78.37 | 4.727 | 8.561 |
| SparseGPT | 57.85 | 82.15 | 80.73 | 81.88 | **80.66** | 76.66 | 5.986 | 9.764 |
| +DenoiseRotator | **61.35** | **84.68** | **83.46** | **83.68** | 79.56 | **78.54** | **4.607** | **8.363** |
| 2:4 Magnitude | 40.44 | 63.01 | 60.51 | 73.39 | 57.85 | 59.04 | 18.169 | 25.919 |
| +DenoiseRotator | 40.19 | 64.27 | 66.05 | 73.99 | 65.19 | 61.94 | 11.528 | 20.073 |
| Wanda | 48.89 | 75.46 | 74.01 | 80.36 | 71.74 | 70.09 | 9.388 | 14.679 |
| +DenoiseRotator | 56.57 | **81.9** | 80.12 | 81.56 | 78.53 | 75.74 | 6.597 | 10.907 |
| SparseGPT | 47.27 | 74.37 | 69.7 | 78.94 | 75.53 | 69.16 | 10.972 | 16.652 |
| +DenoiseRotator | **57.85** | 81.73 | **81.1** | **81.99** | **79.16** | **76.37** | **6.253** | **10.113** |
| 4:8 Magnitude | 44.54 | 67.42 | 66.27 | 75.24 | 62.75 | 63.25 | 12.653 | 19.851 |
| +DenoiseRotator | 45.73 | 73.48 | 60.87 | 75.14 | 66.85 | 64.42 | 10.579 | 18.887 |
| Wanda | 52.9 | 79.71 | 78.55 | 81.94 | 74.43 | 73.51 | 7.158 | 11.746 |
| +DenoiseRotator | 59.13 | 82.91 | 82.04 | 82.26 | 78.37 | 76.94 | 5.714 | 9.709 |
| SparseGPT | 54.44 | 80.26 | 76.51 | 81.12 | 77.43 | 73.95 | 8.011 | 12.736 |
| +DenoiseRotator | **60.75** | **83.59** | **82.52** | **82.81** | **79.32** | **77.8** | **5.453** | **9.175** |

Table 14: Pruning Mistral-7B

| Model | | Zero-shot accuracy (%) | | | | | | Perplexity | |
|---|---|---|---|---|---|---|---|---|---|
| Mistral-7B | | ARC-c | ARC-e | Hellaswag | Piqa | Winogrande | Average | wikitext2 | c4 |
| 0% | Dense | 56.14 | 76.89 | 83.68 | 80.63 | 73.72 | 74.21 | 5.947 | 9.741 |
| 50% | Magnitude | 35.92 | 58.46 | 59.33 | 72.2 | 62.98 | 57.78 | 30.393 | 36.197 |
| | +DenoiseRotator | 51.02 | 70.2 | 77.65 | 78.24 | 71.82 | 69.79 | 7.303 | 11.611 |
| | Wanda | 51.88 | 75.63 | 78.36 | 79.76 | 70.4 | 71.21 | 6.916 | 10.914 |
| | +DenoiseRotator | **54.69** | **76.09** | 80.52 | 79.6 | 72.38 | 72.66 | 6.516 | 10.385 |
| | SparseGPT | 51.96 | 74.33 | 79.53 | 80.47 | 72.53 | 71.76 | 6.938 | 10.783 |
| | +DenoiseRotator | 54.52 | **76.09** | **81.94** | **81.18** | **73.56** | **73.46** | **6.376** | **10.228** |
| 2:4 | Magnitude | 30.03 | 42.68 | 46.25 | 66.38 | 54.85 | 48.04 | 141.962 | 183.124 |
| | +DenoiseRotator | 40.44 | 60.65 | 65.39 | 73.72 | 62.67 | 60.57 | 9.523 | 14.718 |
| | Wanda | 43.6 | 66.16 | 65.79 | 75.73 | 65.11 | 63.28 | 10.176 | 15.572 |
| | +DenoiseRotator | 49.06 | 73.61 | 75.65 | 79.38 | 70.32 | 69.61 | 7.796 | 12.005 |
| | SparseGPT | 46.84 | 70.2 | 69.64 | 76.5 | 67.72 | 66.18 | 9.714 | 13.974 |
| | +DenoiseRotator | **51.02** | **74.58** | **77.38** | **80.09** | **71.82** | **70.98** | **7.301** | **11.229** |
| 4:8 | Magnitude | 32.42 | 49.92 | 53.67 | 70.08 | 59.91 | 53.2 | 47.863 | 59.893 |
| | +DenoiseRotator | 46.25 | 65.53 | 73.34 | 78.18 | 67.56 | 66.17 | 8.061 | 12.678 |
| | Wanda | 49.15 | 72.56 | 72.85 | 77.75 | 69.93 | 68.45 | 8.108 | 12.551 |
| | +DenoiseRotator | 52.9 | 74.54 | 78.77 | 78.84 | **72.06** | 71.42 | 7.056 | 11.019 |
| | SparseGPT | 50.17 | 71.63 | 74.27 | 77.97 | 70.17 | 68.84 | 8.062 | 12.031 |
| | +DenoiseRotator | **53.24** | **74.71** | **79.47** | **80.63** | 71.27 | **71.86** | **6.724** | **10.617** |

Table 15: Pruning Qwen-2.5-7B

| Model | | Zero-shot accuracy (%) | | | | | | Perplexity | |
|---|---|---|---|---|---|---|---|---|---|
| Qwen-2.5-7B | | ARC-c | ARC-e | Hellaswag | Piqa | Winogrande | Average | wikitext2 | c4 |
| 0% | Dense | 51.37 | 77.61 | 78.92 | 80.14 | 72.85 | 72.18 | 6.846 | 11.881 |
| 50% | Magnitude | 26.37 | 40.91 | 30.02 | 50.11 | 48.07 | 39.09 | 198.879 | 217.043 |
| | +DenoiseRotator | 45.56 | 73.19 | 70.75 | 76.99 | 67.64 | 66.83 | 9.272 | 15.702 |
| | Wanda | 44.62 | 71.97 | 71.61 | 77.8 | 70.4 | 67.28 | 8.614 | 14.248 |
| | +DenoiseRotator | 52.47 | **79.38** | 74.7 | 78.84 | 71.35 | 71.35 | 7.932 | 13.277 |
| | SparseGPT | 47.95 | 74.12 | 73.58 | 77.75 | 71.35 | 68.95 | 8.455 | 13.587 |
| | +DenoiseRotator | **53.33** | 79.0 | **76.18** | **79.92** | **72.45** | **72.18** | **7.599** | **12.789** |
| 2:4 | Magnitude | 24.74 | 34.97 | 38.2 | 55.6 | 49.01 | 40.51 | 559.866 | 948.652 |
| | +DenoiseRotator | 42.15 | 69.11 | 63.99 | 74.76 | 62.04 | 62.41 | 11.966 | 19.712 |
| | Wanda | 41.55 | 69.65 | 59.19 | 71.87 | 63.93 | 61.24 | 15.008 | 23.414 |
| | +DenoiseRotator | 48.38 | 74.75 | 68.21 | 77.2 | **70.72** | 67.85 | 10.133 | 16.472 |
| | SparseGPT | 42.92 | 72.05 | 64.3 | 74.86 | 69.22 | 64.67 | 11.349 | 17.179 |
| | +DenoiseRotator | **50.0** | **77.95** | **71.41** | **77.8** | 70.64 | **69.56** | **8.878** | **14.334** |
| 4:8 | Magnitude | 24.49 | 37.88 | 32.81 | 51.47 | 50.28 | 39.38 | 1188.284 | 1095.696 |
| | +DenoiseRotator | 44.88 | 70.41 | 67.54 | 75.84 | 65.98 | 64.93 | 10.379 | 17.446 |
| | Wanda | 45.73 | 73.27 | 67.02 | 75.68 | 69.85 | 66.31 | 10.472 | 16.973 |
| | +DenoiseRotator | **51.19** | **78.54** | 71.53 | 77.31 | 70.48 | 69.81 | 8.832 | 14.612 |
| | SparseGPT | 48.89 | 76.39 | 68.81 | 76.82 | 70.17 | 68.22 | 9.612 | 15.088 |
| | +DenoiseRotator | 50.94 | 76.68 | **73.66** | **78.29** | 70.64 | **70.04** | **8.168** | **13.435** |

Table 16: Pruning Qwen-2.5-14B

| Model | Zero-shot accuracy (%) | | | | | | Perplexity | |
|---|---|---|---|---|---|---|---|---|
| Qwen-2.5-14B | ARC-c | ARC-e | Hellaswag | Piqa | Winogrande | Average | wikitext2 | c4 |
| 0% Dense | 58.96 | 79.34 | 82.95 | 82.1 | 75.69 | 75.81 | 5.294 | 10.349 |
| **50%** Magnitude | 40.36 | 59.81 | 54.07 | 70.4 | 62.98 | 57.52 | 22.936 | 32.148 |
| +DenoiseRotator | 41.21 | 65.11 | 63.34 | 75.9 | 66.69 | 62.45 | 8.781 | 15.919 |
| Wanda | 55.12 | 82.45 | 76.24 | 79.98 | 73.01 | 73.36 | 7.313 | 12.429 |
| +DenoiseRotator | 57.42 | 82.91 | 79.56 | **80.96** | **75.06** | 75.18 | 6.719 | 11.634 |
| SparseGPT | 55.29 | 82.24 | 77.71 | 80.09 | 74.66 | 74.0 | 7.271 | 12.076 |
| +DenoiseRotator | **59.22** | **83.84** | **79.92** | 80.9 | 74.66 | **75.71** | **6.505** | **11.325** |
| **2:4** Magnitude | 36.43 | 56.69 | 49.49 | 69.15 | 57.7 | 53.89 | 58.927 | 68.898 |
| +DenoiseRotator | 38.48 | 67.13 | 55.14 | 71.44 | 62.27 | 58.89 | 13.513 | 22.581 |
| Wanda | 41.38 | 70.12 | 66.15 | 75.19 | 70.48 | 64.66 | 11.658 | 18.569 |
| +DenoiseRotator | 48.63 | 76.56 | 73.83 | 78.02 | 74.11 | 70.23 | 8.705 | 14.419 |
| SparseGPT | 43.43 | 72.18 | 69.08 | 76.82 | 72.3 | 66.76 | 10.196 | 15.678 |
| +DenoiseRotator | **52.9** | **79.34** | **75.84** | **78.78** | **75.06** | **72.38** | **7.858** | **12.875** |
| **4:8** Magnitude | 37.97 | 60.19 | 50.75 | 71.82 | 60.22 | 56.19 | 29.907 | 42.944 |
| +DenoiseRotator | 43.43 | 70.71 | 60.85 | 74.86 | 66.06 | 63.18 | 9.871 | 17.492 |
| Wanda | 49.91 | 78.58 | 72.18 | 77.53 | 72.14 | 70.07 | 8.829 | 14.446 |
| +DenoiseRotator | 55.03 | **82.74** | 77.41 | 79.76 | 74.35 | 73.86 | 7.533 | 12.703 |
| SparseGPT | 48.89 | 78.07 | 73.18 | 78.89 | 73.09 | 70.42 | 8.398 | 13.539 |
| +DenoiseRotator | **55.38** | 81.61 | **78.17** | **80.47** | **74.51** | **74.03** | **7.127** | **11.997** |

Table 17: Pruning Qwen-2.5-32B

| Model | Zero-shot accuracy (%) | | | | | | Perplexity | |
|---|---|---|---|---|---|---|---|---|
| Qwen-2.5-32B | ARC-c | ARC-e | Hellaswag | Piqa | Winogrande | Average | wikitext2 | c4 |
| 0% Dense | 55.72 | 78.03 | 84.12 | 82.05 | 75.69 | 75.12 | 5.018 | 10.169 |
| **50%** Magnitude | 46.93 | 66.62 | 63.1 | 72.96 | 69.46 | 63.81 | 19.221 | 33.582 |
| +DenoiseRotator | 47.7 | 73.7 | 73.63 | 79.43 | 71.51 | 69.19 | 6.971 | 12.845 |
| Wanda | 54.44 | 77.36 | 80.63 | 81.45 | 75.22 | 73.82 | 6.297 | 11.359 |
| +DenoiseRotator | **58.62** | **83.38** | 82.34 | **82.15** | 75.45 | **76.39** | 5.993 | 10.998 |
| SparseGPT | 53.5 | 78.62 | 81.25 | 81.34 | 75.61 | 74.06 | 6.348 | 11.179 |
| +DenoiseRotator | 57.76 | 81.69 | **82.56** | 81.94 | **76.16** | 76.02 | **5.858** | **10.76** |
| **2:4** Magnitude | 43.6 | 62.42 | 57.82 | 69.75 | 65.35 | 59.79 | 24.272 | 45.718 |
| +DenoiseRotator | 46.08 | 71.17 | 68.25 | 78.07 | 68.67 | 66.45 | 8.613 | 14.929 |
| Wanda | 48.89 | 76.64 | 75.52 | 79.65 | 74.74 | 71.09 | 8.077 | 13.819 |
| +DenoiseRotator | **53.41** | **80.51** | 78.02 | 79.6 | 75.3 | **73.37** | 7.901 | 13.632 |
| SparseGPT | 48.63 | 74.62 | 76.45 | 79.92 | **75.85** | 71.1 | 7.919 | 13.047 |
| +DenoiseRotator | 51.88 | 78.32 | **79.74** | **80.69** | 75.61 | 73.25 | **6.754** | **11.695** |
| **4:8** Magnitude | 45.22 | 65.28 | 61.51 | 72.03 | 68.43 | 62.5 | 21.363 | 36.348 |
| +DenoiseRotator | 46.84 | 72.22 | 71.65 | 78.4 | 69.22 | 67.67 | 7.619 | 13.565 |
| Wanda | 53.92 | 79.42 | 78.25 | 80.2 | 75.69 | 73.5 | 7.073 | 12.314 |
| +DenoiseRotator | **57.0** | **81.06** | 80.37 | **81.56** | **77.03** | **75.4** | 6.619 | 11.696 |
| SparseGPT | 52.56 | 79.17 | 78.63 | 80.52 | 76.32 | 73.44 | 7.099 | 12.015 |
| +DenoiseRotator | 55.03 | **81.06** | **81.42** | **81.56** | 75.61 | 74.94 | **6.324** | **11.177** |

Table 18: Pruning Qwen-2.5-72B

| Model | | Zero-shot accuracy (%) | | | | | | Perplexity | |
| Qwen-2.5-72B | | ARC-c | ARC-e | Hellaswag | Piqa | Winogrande | Average | wikitext2 | c4 |
|---|---|---|---|---|---|---|---|---|---|
| 0% | Dense | 62.46 | 83.42 | 86.01 | 83.73 | 77.82 | 78.69 | 3.875 | 9.26 |
| 50% | Magnitude | 30.46 | 46.17 | 28.73 | 47.33 | 53.67 | 41.27 | 734.042 | 557.275 |
| | +DenoiseRotator | 57.08 | 81.02 | 80.83 | 82.37 | 76.4 | 75.54 | 5.369 | 10.695 |
| | Wanda | 60.92 | 84.68 | 83.5 | 82.43 | 78.69 | 78.04 | 5.218 | 10.229 |
| | +DenoiseRotator | 61.86 | 83.46 | 84.42 | **83.19** | 78.69 | 78.32 | 4.937 | 9.956 |
| | SparseGPT | 61.52 | **85.19** | 83.29 | 82.15 | **79.08** | 78.25 | 4.937 | 9.957 |
| | +DenoiseRotator | **62.2** | 84.51 | **84.8** | 83.08 | 77.51 | **78.42** | **4.778** | **9.761** |
| 2:4 | Magnitude | 28.75 | 47.01 | 28.32 | 49.67 | 53.28 | 41.41 | 287.701 | 314.286 |
| | +DenoiseRotator | 50.43 | 74.41 | 72.32 | 79.22 | 70.72 | 69.42 | 8.81 | 14.647 |
| | Wanda | 56.66 | 81.02 | 79.01 | 81.61 | 76.4 | 74.94 | 6.692 | 11.883 |
| | +DenoiseRotator | **60.75** | **84.81** | 81.96 | 82.1 | **78.69** | **77.66** | 6.164 | 11.258 |
| | SparseGPT | 56.31 | 82.41 | 78.59 | 81.28 | 78.53 | 75.43 | 7.187 | 11.947 |
| | +DenoiseRotator | 59.81 | 84.22 | **82.29** | **82.26** | 77.19 | 77.16 | **5.851** | **10.715** |
| 4:8 | Magnitude | 30.29 | 47.14 | 25.38 | 48.04 | 52.01 | 40.57 | 456.802 | 397.805 |
| | +DenoiseRotator | 52.39 | 76.35 | 78.62 | 82.05 | 74.03 | 72.69 | 6.138 | 11.434 |
| | Wanda | 60.92 | **85.4** | 81.45 | 82.48 | 77.43 | 77.53 | 5.939 | 10.938 |
| | +DenoiseRotator | 59.64 | 83.5 | 83.45 | **83.08** | 78.14 | 77.56 | 5.511 | 10.538 |
| | SparseGPT | **61.09** | 84.72 | 81.33 | 82.32 | **79.79** | **77.85** | 6.158 | 10.962 |
| | +DenoiseRotator | 58.19 | 83.71 | **83.81** | 82.48 | 76.16 | 76.87 | **5.295** | **10.164** |

Table 19: Pruning Llama-2-7B

| Model | | Zero-shot accuracy (%) | | | | | | Perplexity | |
| Llama-2-7B | | ARC-c | ARC-e | Hellaswag | Piqa | Winogrande | Average | wikitext2 | c4 |
|---|---|---|---|---|---|---|---|---|---|
| 0% | Dense | 46.25 | 74.54 | 75.98 | 79.11 | 69.14 | 69.0 | 5.472 | 7.263 |
| 50% | Wanda | 42.83 | 68.69 | 70.65 | 76.88 | 66.85 | 65.18 | 6.9 | 9.231 |
| | +DenoiseRotator | 42.83 | 69.95 | 71.83 | **77.2** | 68.11 | 65.98 | 6.519 | 8.817 |
| | SparseGPT | 41.3 | 67.38 | 71.11 | 77.04 | **69.3** | 65.22 | 7.003 | 9.254 |
| | +DenoiseRotator | **42.92** | **72.26** | **72.09** | 76.99 | 68.59 | **66.56** | **6.273** | **8.409** |
| 2:4 | Wanda | 31.57 | 57.11 | 55.13 | 71.22 | 62.59 | 55.52 | 12.265 | 15.856 |
| | +DenoiseRotator | 34.47 | 62.84 | 61.79 | **74.43** | 65.67 | 59.83 | 9.876 | 12.99 |
| | SparseGPT | 33.36 | 59.05 | 57.99 | 71.76 | 65.43 | 57.51 | 11.291 | 14.301 |
| | +DenoiseRotator | **37.37** | **64.98** | **64.64** | 74.37 | **66.61** | **61.59** | **8.696** | **11.156** |
| 4:8 | Wanda | 37.71 | 63.51 | 64.32 | 74.59 | 67.01 | 61.42 | 8.613 | 11.418 |
| | +DenoiseRotator | 38.23 | 65.28 | 66.92 | 75.41 | 66.61 | 62.48 | 7.645 | 10.286 |
| | SparseGPT | **38.57** | 63.17 | 65.19 | 75.41 | 67.48 | 61.96 | 8.689 | 10.946 |
| | +DenoiseRotator | 37.29 | **65.61** | **68.46** | **76.55** | **68.59** | **63.3** | **7.157** | **9.543** |

Table 20: Pruning Llama-2-13B

| Model | | | | | | | | Perplexity | |
|---|---|---|---|---|---|---|---|---|---|
| | Llama-2-13B | ARC-c | ARC-e | Hellaswag | Piqa | Winogrande | Average | wikitext2 | c4 |
| 0% | Dense | 49.15 | 77.48 | 79.37 | 80.52 | 72.14 | 71.73 | 4.883 | 6.727 |
| 50% | Wanda | 46.5 | 70.66 | **76.07** | 79.38 | 71.43 | 68.8 | 5.965 | 8.297 |
| | +DenoiseRotator | **47.53** | 73.48 | 75.52 | 78.56 | **71.82** | 69.38 | 5.779 | 7.957 |
| | SparseGPT | 45.73 | 71.25 | 75.16 | 79.33 | 71.11 | 68.51 | 6.058 | 8.261 |
| | +DenoiseRotator | 47.44 | **75.72** | 75.94 | 79.0 | 70.72 | **69.76** | **5.515** | **7.582** |
| 2:4 | Wanda | 37.54 | 64.35 | 62.54 | 75.52 | 67.4 | 61.47 | 9.036 | 12.564 |
| | +DenoiseRotator | 38.14 | 63.34 | 64.92 | 73.07 | 68.19 | 61.53 | 8.772 | 12.46 |
| | SparseGPT | 38.48 | 63.8 | 64.44 | 75.68 | 69.46 | 62.37 | 9.058 | 11.804 |
| | +DenoiseRotator | **41.64** | **70.66** | **70.77** | **76.17** | **72.06** | **66.26** | **6.811** | **9.322** |
| 4:8 | Wanda | 43.17 | 68.35 | 70.88 | 77.31 | 69.3 | 65.8 | 6.96 | 9.72 |
| | +DenoiseRotator | 43.0 | **71.68** | 70.9 | 76.77 | 71.27 | 66.72 | 6.999 | 10.041 |
| | SparseGPT | 43.09 | 69.07 | 70.48 | 76.71 | 70.56 | 65.98 | 7.222 | 9.705 |
| | +DenoiseRotator | **43.6** | 71.46 | **73.95** | **78.02** | **72.53** | **67.91** | **6.119** | **8.335** |

Table 21: Pruning Llama-2-70B

| Model | | | | | | | | Perplexity | |
|---|---|---|---|---|---|---|---|---|---|
| | Llama-2-70B | ARC-c | ARC-e | Hellaswag | Piqa | Winogrande | Average | wikitext2 | c4 |
| 0% | Dense | 57.34 | 80.98 | 83.84 | 82.7 | 77.98 | 76.56 | 3.319 | 5.709 |
| 50% | Wanda | **55.63** | 78.91 | 81.28 | **82.59** | 77.74 | 75.23 | 4.218 | 6.5 |
| | +DenoiseRotator | 55.55 | **80.22** | 82.01 | 82.48 | 77.51 | **75.55** | 3.894 | 6.174 |
| | SparseGPT | 55.55 | 79.67 | 81.27 | 82.21 | **78.14** | 75.36 | 4.256 | 6.459 |
| | +DenoiseRotator | 55.2 | 79.5 | **82.45** | 82.48 | 77.03 | 75.33 | **3.81** | **6.075** |
| 2:4 | Wanda | 50.85 | 77.44 | 75.96 | 79.76 | 75.37 | 71.87 | 5.468 | 8.118 |
| | +DenoiseRotator | 52.9 | 78.03 | **79.82** | 80.74 | **77.66** | 73.83 | 4.845 | 7.231 |
| | SparseGPT | 50.68 | 76.77 | 75.9 | 79.16 | 75.22 | 71.54 | 5.727 | 8.16 |
| | +DenoiseRotator | **53.16** | **78.32** | 79.48 | **81.28** | 77.27 | **73.9** | **4.643** | **6.936** |
| 4:8 | Wanda | 53.24 | 79.0 | 78.77 | 81.18 | 75.93 | 73.62 | 4.764 | 7.156 |
| | +DenoiseRotator | **54.18** | 79.0 | 80.97 | **81.72** | 76.87 | 74.54 | 4.333 | 6.62 |
| | SparseGPT | 53.84 | 78.7 | 78.68 | 80.79 | 76.56 | 73.71 | 4.926 | 7.201 |
| | +DenoiseRotator | 54.01 | **79.5** | **81.41** | 81.61 | **77.03** | **74.71** | **4.247** | **6.467** |

Table 22: Pruning Llama-1-7B

| Model | | | | | | | | Perplexity | |
|---|---|---|---|---|---|---|---|---|---|
| | Llama-1-7B | ARC-c | ARC-e | Hellaswag | Piqa | Winogrande | Average | wikitext2 | c4 |
| 0% | Dense | 44.62 | 72.94 | 76.22 | 79.16 | 69.93 | 68.57 | 5.677 | 7.343 |
| 50% | Wanda | 40.61 | 64.65 | 69.96 | 77.42 | 66.46 | 63.81 | 7.243 | 9.324 |
| | +DenoiseRotator | **41.47** | **69.23** | 71.74 | **78.35** | 68.59 | 65.87 | 6.553 | 8.599 |
| | SparseGPT | 39.85 | 65.28 | 69.47 | 77.48 | 69.22 | 64.25 | 7.247 | 9.347 |
| | +DenoiseRotator | 41.3 | **69.23** | **72.02** | 77.26 | **70.17** | **65.99** | **6.379** | **8.317** |
| 2:4 | Wanda | 31.31 | 55.01 | 56.63 | 70.73 | 62.12 | 55.15 | 11.613 | 14.668 |
| | +DenoiseRotator | 34.3 | **60.98** | 62.78 | **75.35** | **68.67** | 60.41 | 9.214 | 11.899 |
| | SparseGPT | 32.51 | 56.78 | 57.7 | 72.2 | 65.11 | 56.85 | 11.341 | 13.847 |
| | +DenoiseRotator | **35.41** | 60.56 | **65.54** | 74.48 | 67.09 | **60.61** | **8.233** | **10.489** |
| 4:8 | Wanda | 34.39 | 59.68 | 63.55 | 74.21 | 64.33 | 59.23 | 8.595 | 11.268 |
| | +DenoiseRotator | 37.12 | 63.68 | 67.45 | 74.65 | **68.82** | 62.34 | 7.506 | 9.868 |
| | SparseGPT | 35.32 | 59.85 | 64.26 | 74.37 | 65.51 | 59.86 | 8.67 | 10.914 |
| | +DenoiseRotator | **39.25** | **65.28** | **68.78** | **76.66** | 68.67 | **63.72** | **7.136** | **9.254** |

Table 23: Pruning Llama-1-13B

| Model | | Zero-shot accuracy (%) | | | | | | Perplexity | |
|---|---|---|---|---|---|---|---|---|---|
| | Llama-1-13B | ARC-c | ARC-e | Hellaswag | Piqa | Winogrande | Average | wikitext2 | c4 |
| 0% | Dense | 47.61 | 74.75 | 79.04 | 80.2 | 72.69 | 70.85 | 5.09 | 6.798 |
| 50% | Wanda | 43.86 | 70.33 | 74.75 | 78.67 | 71.43 | 67.8 | 6.134 | 8.134 |
| | +DenoiseRotator | 45.73 | **72.43** | 76.18 | 78.89 | 71.59 | 68.96 | **5.663** | 7.595 |
| | SparseGPT | 40.61 | 66.2 | 74.25 | 78.35 | 72.06 | 66.29 | 6.229 | 8.145 |
| | +DenoiseRotator | **46.16** | 71.46 | **76.62** | **79.16** | **73.32** | **69.34** | 5.665 | **7.458** |
| 2:4 | Wanda | 36.43 | 60.61 | 63.66 | 73.88 | 68.43 | 60.6 | 9.58 | 12.125 |
| | +DenoiseRotator | 39.76 | 66.33 | 67.86 | 76.12 | 69.61 | 63.93 | 7.631 | 12.271 |
| | SparseGPT | 36.09 | 60.82 | 65.6 | 75.19 | 69.22 | 61.38 | 9.084 | 11.385 |
| | +DenoiseRotator | **42.32** | **69.44** | **71.16** | **76.71** | **71.19** | **66.16** | **6.902** | **8.993** |
| 4:8 | Wanda | 40.78 | 66.37 | 69.75 | 77.04 | 71.35 | 65.05 | 7.384 | 9.538 |
| | +DenoiseRotator | 42.41 | 70.66 | 72.98 | 77.8 | 70.96 | 66.96 | 6.383 | 8.538 |
| | SparseGPT | 41.21 | 67.68 | 69.75 | 76.61 | 71.43 | 65.33 | 7.513 | 9.458 |
| | +DenoiseRotator | **43.6** | **70.83** | **73.99** | **78.67** | **73.16** | **68.05** | **6.221** | **8.146** |

Table 24: Pruning Llama-1-30B

| Model | | Zero-shot accuracy (%) | | | | | | Perplexity | |
|---|---|---|---|---|---|---|---|---|---|
| | Llama-1-30B | ARC-c | ARC-e | Hellaswag | Piqa | Winogrande | Average | wikitext2 | c4 |
| 0% | Dense | 52.99 | 78.91 | 82.65 | 82.21 | 75.77 | 74.5 | 4.1 | 6.129 |
| 50% | Wanda | **51.79** | **77.78** | 79.97 | 79.87 | 73.09 | **72.49** | 5.242 | 7.274 |
| | +DenoiseRotator | 49.83 | 77.36 | 79.82 | 80.03 | **75.06** | 72.42 | 4.815 | 6.823 |
| | SparseGPT | 50.85 | 76.77 | 79.27 | 80.47 | 74.43 | 72.35 | 5.35 | 7.351 |
| | +DenoiseRotator | 49.57 | 75.67 | **80.34** | **81.12** | 73.88 | 72.11 | **4.7** | **6.683** |
| 2:4 | Wanda | **45.31** | 71.17 | 73.32 | 78.56 | 71.35 | 67.94 | 6.929 | 9.505 |
| | +DenoiseRotator | 44.54 | 72.73 | 75.17 | 78.24 | **75.53** | 69.24 | 6.17 | 8.442 |
| | SparseGPT | 43.94 | 72.1 | 72.61 | 78.24 | 71.9 | 67.75 | 7.244 | 9.542 |
| | +DenoiseRotator | **45.31** | **74.07** | **76.21** | **79.11** | 73.72 | **69.68** | **5.822** | **7.888** |
| 4:8 | Wanda | 49.06 | 75.38 | 76.58 | 79.38 | 72.93 | 70.66 | 5.967 | 8.162 |
| | +DenoiseRotator | **49.23** | 73.74 | 77.83 | 79.38 | 75.06 | 71.04 | 5.467 | 7.482 |
| | SparseGPT | 48.12 | 75.17 | 75.52 | 79.71 | 73.72 | 70.44 | 6.239 | 8.26 |
| | +DenoiseRotator | 48.89 | **76.43** | **78.26** | **80.58** | **75.85** | **72.0** | **5.267** | **7.216** |

Table 25: Pruning Llama-1-65B

| Model | | Zero-shot accuracy (%) | | | | | | Perplexity | |
|---|---|---|---|---|---|---|---|---|---|
| | Llama-1-65B | ARC-c | ARC-e | Hellaswag | Piqa | Winogrande | Average | wikitext2 | c4 |
| 0% | Dense | 55.63 | 79.76 | 84.13 | 82.26 | 77.35 | 75.82 | 3.532 | 5.811 |
| 50% | Wanda | 53.58 | 77.78 | 81.98 | **81.94** | 76.8 | 74.41 | 4.616 | 6.694 |
| | +DenoiseRotator | 53.84 | 78.03 | 81.4 | 81.39 | 76.01 | 74.13 | 4.223 | 6.347 |
| | SparseGPT | 54.18 | 76.56 | 81.89 | 81.72 | **77.19** | 74.3 | 4.601 | 6.661 |
| | +DenoiseRotator | **54.78** | **79.76** | **82.43** | 81.83 | 77.11 | **75.18** | **4.094** | **6.229** |
| 2:4 | Wanda | 46.84 | 75.17 | 76.04 | 79.16 | 75.3 | 70.5 | 6.234 | 8.829 |
| | +DenoiseRotator | 47.01 | 74.12 | 77.4 | 78.73 | 74.98 | 70.44 | 5.584 | 7.776 |
| | SparseGPT | 48.81 | 75.63 | 76.11 | 79.49 | 76.95 | 71.39 | 6.263 | 8.427 |
| | +DenoiseRotator | **51.96** | **76.64** | **79.48** | **79.76** | **77.82** | **73.13** | **5.061** | **7.222** |
| 4:8 | Wanda | 51.11 | 77.4 | 79.04 | **81.23** | 76.56 | 73.06 | 5.297 | 7.506 |
| | +DenoiseRotator | 51.45 | **78.03** | 80.45 | 79.92 | 75.93 | 73.15 | 4.741 | 6.871 |
| | SparseGPT | 51.19 | 76.22 | 79.04 | 80.47 | **77.66** | 72.91 | 5.308 | 7.404 |
| | +DenoiseRotator | **52.99** | 77.78 | **80.58** | **81.23** | 77.03 | **73.92** | **4.558** | **6.689** |

