# OpenReview forum: "DenoiseRotator: Enhance Pruning Robustness for LLMs via Importance Concentration"
_NeurIPS.cc/2025/Conference — NeurIPS 2025 poster_

### Official Review · Reviewer_PoQx · 2025-06-30

**Clarity:** 3
**Significance:** 3
**Originality:** 3
**Rating:** 4
**Confidence:** 4

**Summary:**

The paper introduces DenoiseRotator, a plug‑and‑play pre‑pruning procedure for large language models (LLMs). Instead of only selecting low‑importance weights, the main contribution of this paper is to introduce an importance score redistribution framework by inserting two pairs of learnable orthogonal matrices per Transformer block. These matrices are trained to minimise the information entropy of the normalised importance scores, so that the importance scores concentrate on fewer weights, making any subsequent pruning step more robust. Extensive experiments on Mistral‑7B, LLaMA‑3 (8 B & 70 B), and Qwen‑2.5 (up to 72 B) show large perplexity and zero‑shot‑accuracy gains at both 50 % unstructured and semi-structured sparsity compared with the underlying pruners.

**Questions:**

- Parameter Overhead Analysis. For 70 B models, the rotations add >7 B parameters (~10 %) before merging. Memory spikes during training might be a concern. Could low‑rank or block‑diagonal orthogonals work? An ablation on rank‑k Householder products would clarify the trade‑off. I noticed you discussed it in lines 232-234, but a more detailed elaboration regarding the training timing and memory costs would be appreciated.

- Baseline breadth. Your work's high-level idea is quite similar to RotPruner. Would you please provide a detailed comparison of these two works? It would also be appreciated if the performance of RotPruner is included in the baseline.

- Downstream fine-tuning compatibility. How would you recommend handling $R_1/R_1^\top$ before LoRA/4-bit quant? A short empirical check (e.g., 1-epoch LoRA on 8 B) would reassure users.

- Structured-mask awareness. In lines 245-248, you mentioned DenoiseRotator works remarkably well for 2:4 sparsity even without an explicit constraint. Have you tried explicitly learning rotations under a 2:4 mask, e.g., adding a differentiable surrogate for block occupancy? Results suggest good luck, but a principled variant might push performance further.

- Clarify Fig. 2 (c) (d). Why those low-importance weights are unpruned? Did you mislabel?

**Ethical Concerns:**

["NO or VERY MINOR ethics concerns only"]

**Final Justification:**

I think all my concerns have been addressed and I would like to maintain my current recommendations.

**Limitations:**

Partially addressed.
The paper notes that semi-structured masks impose constraints not yet modelled, but omits:
- Peak memory and time cost of rotation training on very large models.
- Possible risks when further fine-tuning a model that still contains one orthogonal pair per block.

Adding brief quantitative statements on these points would make the limitations section complete.

**Quality:**

3

**Strengths And Weaknesses:**

## Strengths
### Quality
- This paper is technically sound, with a clear mathematical formulation linking entropy to pruning robustness.
- Solid implementation details (QR re‑param, invariance proof).
- Large-scale evaluation across three types of models, various sizes, three baselines of importance scoring methods, and three sparsity patterns.

### Clarity
- Writing is generally good. Organisation and notation are easy to follow. Fig. 1 & Alg. 1 are helpful.
- Appendix lists transformed score formulas, aiding reproducibility.

### Significance
- Clear improvement over baselines at both 50 %, 2 : 4 and 4:8 masks.
- The method is agnostic to the pruner; practitioners can add it cheaply before any favourite pruning tool.

### Originality
- Introduces entropy-minimised importance shaping, which is new in the pruning literature.

## Weaknesses
### Quality
- Recovery /downstreaming fine-tuning (FT) is not discussed. Since the pruned model still contains one orthogonal pair per block, discussing the potential risk/instability of light FT would be beneficial.

### Clarity
- Some images are confusing (e.g., Fig. 2 (c) & (d)). See Questions.

### Significance
- Some closely related baselines are not included. See Questions.

### Originality
- High-level idea (“learn orthogonal rotations, then prune”) overlaps with RotPruner [1]. Without a head-to-head, the novelty margin is hard to judge.

[1] Haoxian Chen and Limin Wang. Rotpruner: Large language model pruning in rotated space, 2025.

---

> ### Author Rebuttal · Authors · 2025-07-29
>
> Thank you for your comprehensive review and constructive feedback on our paper. We are grateful for your recognition of the technical soundness and originality of our work, DenoiseRotator, which introduces an innovative importance score redistribution framework for enhancing the robustness of pruning in large language models. Your acknowledgment of the clarity in our writing and the significant improvements demonstrated through extensive experiments is highly encouraging.
>
> We understand the concerns raised regarding recovery and downstream fine-tuning, as well as the need for clearer images and a broader baseline comparison, particularly with RotPruner. Additionally, your questions about parameter overhead, downstream fine-tuning compatibility, and structured-mask awareness are crucial for further refining our approach.
>
> Your feedback is invaluable in helping us address these points and improve the clarity and depth of our work. We are committed to providing detailed responses and insights to the questions and weaknesses you've highlighted. Thank you again for your thoughtful comments, which guide us in enhancing the presentation and impact of our research.
>
> ### Question 1:
>
> Thank you for your insightful comments!
>
> We would like to clarify that the training resources required by DenoiseRotator are minimal, substantially less than those needed for actual deployment. Therefore, large language models capable of operating in resource-constrained or deployment-limited environments can also utilize DenoiseRotator for pruning within the same settings.
>
> Due to the independence of rotation matrices across layers, the training of each layer's rotation matrix can be conducted separately. The actual necessary memory consumption is calculated as 16 (parameters 2 + gradients 4 + optimizer states 8) times hidden times hidden, equivalent to the memory required for deploying one layer. In our straightforward, uncompiled implementation, optimizing rotation matrices for LLaMA-3-8B requires only 10GB of memory, while LLaMA-3-70B requires only 30GB.
>
> Regarding the use of low-rank matrices for orthogonal transformations, it is important to note that orthogonal matrices must be full-rank square matrices. Thus, matrices derived from low-rank approximations cannot maintain orthogonality, and due to the presence of RMSNorm, they cannot be merged into the model's weights.
>
> Rank-k Householder products can only reduce the computational complexity during training but do not reduce the inference overhead.
>
> Block-diagonal matrices offer a promising trade-off by reducing both training and inference overhead. To validate their effectiveness, we conducted experiments on LLaMA-3-8B:
>
> Setup: SparseGPT, unstructured 50%, 2000 steps, learning rate 0.01
>
> | Block Number | Perplexity | Zero-Shot Accuracy | Average Time Cost per Step (s) | Entropy |
> |----|---|-----|---|--|
> | 1   | 7.597  | 69.58   | 0.124   | 384128  |
> | 2   | 8.024  | 68.68| 0.088   | 410816  |
> | 4  | 8.544   | 68.47   | 0.076  | 428160  |
> | 8  | 8.882  | 67.51   | 0.072   | 440512  |
>
> These results demonstrate the trade-offs in performance and computational cost associated with different block configurations. Thank you for your feedback, which has guided us in exploring these options further.
>
> ### Question 2:
>
> The core contribution of our paper lies in the concept of "enhancing model robustness to pruning by concentrating importance onto a smaller subset of parameters." Learning orthogonal rotations followed by pruning is one specific implementation of this idea, tailored to post-train pruning, but it is not the only approach. Additionally, although not experimentally verified in our paper, we propose the notion of incorporating the information entropy of importance scores as a regularization term in the objective function for sparse pre-training. RotPruner, on the other hand, focuses solely on the implementation steps of learning orthogonal rotations followed by pruning without providing theoretical motivation.
>
> Regarding the differences in implementation:
> In the RotPruner framework, the training of rotation matrices (continuous) and parameter pruning (discontinuous) are not independent but alternate. Due to the presence of pruning masks, pruned parameters do not participate in the training of rotation matrices, leading to potential information loss. This alternating implementation method is incompatible with pruners that incorporate parameter reconstruction mechanisms, such as SparseGPT. Moreover, RotPruner requires end-to-end optimization of the entire model, simultaneously optimizing rotation matrices for all layers, resulting in high memory requirements. Furthermore, RotPruner adds two non-mergeable rotation matrices per layer—one for self-attention and one for MLP—adding additional overhead.
>
> In contrast, our proposed DenoiseRotator decouples the training of rotation matrices from parameter pruning. This approach avoids information loss due to discontinuous pruning and offers excellent compatibility with many existing pruners. DenoiseRotator adds only one non-mergeable rotation matrix per layer, and as mentioned in the response to Question 1, the training of each layer's rotation matrix can be conducted independently, significantly reducing the memory required for training.
>
> Since RotPruner has not released their code and has only conducted experiments on models of 8B and below, we provide a comparison using their paper's configuration and data for reference. The calibration dataset consists of 128 samples, each with a length of 2048, sourced from WikiText2. The models' performances are evaluated based on perplexity on the WikiText2 test set. For the optimization of the rotation matrix in DenoiseRotator, the configuration includes a learning rate of 0.01 and 2000 steps.:
>
> Unstructured 50% Sparsity:
> | Method  | LLaMA-2-7B | LLaMA-3-8B |
> |-----|----|----|
> | Dense | 5.47| 6.13  |
> | SparseGPT | 6.46  | 8.29       |
> | SparseGPT + DenoiseRotator | 6.09   | 7.25       |
> | Wanda  | 6.72| 9.40|
> | Wanda + DenoiseRotator  | 6.27       | 7.56       |
> | RotPruner | 6.42  | 8.50  |
>
> Semi-Structured 2:4 Sparsity:
> | Method| LLaMA-2-7B | LLaMA-3-8B |
> |----|---|-----|
> | Dense | 5.47   | 6.13|
> | SparseGPT | 10.37 | 14.65 |
> | SparseGPT + DenoiseRotator | 7.81  | 8.84 |
> | Wanda | 11.34 | 21.21 |
> | Wanda + DenoiseRotator  | 9.37 | 10.68|
> | RotPruner | 9.20 | 11.65 |
>
> DenoiseRotator achieved superior performance with reduced overhead. We hope these comparisons provide clarity on the distinctions and advantages of our approach. Thank you for your feedback, which has guided us in presenting a more comprehensive analysis.
>
> ### Question 3:
> Thank you for your question regarding downstream fine-tuning compatibility, specifically in relation to handling before LoRA/4-bit quantization.
>
> Since quantization fixes the model's weights, it is essential that the training and merging of rotation matrices occur before quantization. Additionally, as pruning removes some of the model's weights, inevitably causing information loss, LoRA should be applied before pruning to preserve as much information as possible.
>
> During training, freezing the rotation matrices can be beneficial. This approach ensures that the gradient flow remains consistent with scenarios where rotations are not applied, thus maintaining training stability.
>
> **DenoiseRotator Configuration:**
> - Steps: 100
> - Learning Rate: 0.01
> - Model: LLaMA-3-8B
> - Pruning: Wanda, 2:4 semi-structured
>
> **Fine-tuning Configuration:**
> - Method: LoRA
> - Alpha: 32.0
> - Dropout: 0.1
> - LoRA-r: 8
> - Dataset: 4096 samples of length 2048 from WikiText2 train set
> - Learning Rate: 2e-4
> - Weight Decay: 1e-2
> - Optimizer: Adam
> - Learning Rate Scheduler: Linear
> - Warm-up Steps: 400
> - Note: Rotation matrices are not fine-tuned
>
> **Results on WikiText2 test set:**
> |Method| Perplexity| Zero-Shot Accuracy |
> |---|---|------|
> | Dense | 6.14  | 72.72  |
> | Wanda | 25.19  | 51.03  |
> | + DenoiseRotator | 14.31 | 60.67   |
> | + LoRA Finetune  | 9.32  | 61.08   |
>
> The experiment demonstrates that applying DenoiseRotator improves performance compared to Wanda alone when considering both perplexity and zero-shot accuracy. Additional recovery finetuning further enhances the model, significantly reducing the perplexity and improving zero-shot accuracy.
>
> ### Question 4:
> Thank you for your insightful suggestion, which highlights an exciting direction for future research!
>
> We have experimented with some straightforward strategies to explicitly incorporate the constraints of 2:4 semi-structured pruning into the optimization of rotation matrices. For instance, we tried replacing the normalization groups used for entropy calculation from entire columns or rows to groups of four consecutive parameters. However, the results were not favorable; perplexity on WikiText2 increased from 11.41 to 17.53 (LLaMA-3-8B, Wanda). We suspect this is due to the focus on local information at the expense of global context.
>
> Subsequently, we attempted to incorporate global information by multiplying the entropy of each normalization group by the sum of the importance scores within that group. This approach improved performance to a perplexity of 14.96, but it still fell short of the method that does not explicitly integrate the constraints of 2:4 semi-structured pruning into the optimization of rotation matrices.
>
> This remains a very promising idea worth exploring further.
>
> ### Question 5:
> Thank you for your observation regarding Figure 2(c) and 2(d). The phenomenon you noted is due to the comparison group used by Wanda, which considers only a single column of the weight matrix. Figures 2(c) and 2(d) depict the statistical analysis results of all parameters contained within the weight matrix. The importance of parameters retained by Wanda within a single column of the weight matrix may not necessarily rank among the highest when considering the importance of all parameters.

---

> > ### Author Response · Authors · 2025-08-04
> > **Looking forward for further feedback**
> >
> > Dear Reviewer PoQx,
> >
> > Thank you once again for your thoughtful and encouraging comments on our submission!
> >
> > We have carefully addressed most of your concerns and are grateful for your valuable suggestions. We will incorporate all of these improvements in the revision.
> >
> > We are actively participating in the Author–Reviewer Discussion phase and would be happy to clarify or elaborate further on any aspect of our work.
> >
> > We look forward to your response!
> >
> > Best regards,
> >
> > Authors of Paper 9077

---

> > ### Comment · Reviewer_PoQx · 2025-08-05
> >
> > Thank you for the detailed and thoughtful response. I appreciate the additional experiments and clarifications. I hope the authors can integrate the above into the revised manuscript. I would like to maintain my current recommendation.

---

### Official Review · Reviewer_3pAp · 2025-07-02

**Clarity:** 3
**Significance:** 3
**Originality:** 2
**Rating:** 4
**Confidence:** 3

**Summary:**

This paper introduces DenoiseRotator, a novel framework that enhances the robustness of neural network pruning for large language models (LLMs) by concentrating parameter importance prior to pruning. Unlike traditional pruning methods that focus solely on selecting which weights to remove, DenoiseRotator takes a fundamentally different approach by first redistributing parameter importance to make models inherently more amenable to pruning. The core innovation lies in applying learnable orthogonal transformations to weight matrices while minimizing the information entropy of normalized importance scores, effectively concentrating importance onto a smaller subset of parameters. The method leverages the computational invariance property of Transformer architectures and is designed to be model-agnostic, allowing seamless integration with existing pruning techniques such as Magnitude, SparseGPT, and Wanda.

**Questions:**

See weakness.

**Ethical Concerns:**

["NO or VERY MINOR ethics concerns only"]

**Final Justification:**

My concerns regarding the entropy-based reasoning have been fully addressed, and my questions about QR decomposition versus Cayley SGD were clearly resolved. Based on these clarifications, I have updated my score to 4.

**Limitations:**

Yes. The paper addressed limitations in the "Limitations" section.

**Quality:**

3

**Strengths And Weaknesses:**

**Strengths**
1. The paper presents a genuinely novel perspective on pruning by focusing on importance redistribution rather than weight selection. The entropy-guided importance concentration approach is theoretically well-motivated and represents a significant departure from existing paradigms.

2. The framework's design allows easy integration with existing pruning pipelines without requiring modifications to the underlying pruning algorithms, enhancing its practical utility.

3. The paper cleverly employs QR decomposition reparameterization to train orthogonal matrices, which elegantly solves the challenge of maintaining orthogonality constraints during gradient-based optimization.

**Weaknesses**
1. Unlike quantization, pruning intuitively targets weights with low importance scores as the primary candidates for removal. Therefore, a transformation that induces a non-uniform distribution (where some weights are naturally pushed toward smaller magnitudes) would seem more appropriate for pruning. However, it is not immediately clear why enforcing a more uniform distribution via entropy minimization would yield better pruning performance. A more thorough explanation is needed to clarify how lower-entropy weight distributions, which typically concentrate probability mass on a subset of values, influence the effectiveness of pruning.

2. While the paper presents QR decomposition as a solution for training orthogonal matrices, it does not clearly explain the specific training advantages this approach offers compared to alternative methods. For instance, SpinQuant[1] demonstrates that Cayley SGD can also effectively maintain orthogonality during backpropagation for rotation matrix optimization. The paper lacks a comparative analysis of why QR decomposition is preferable to other orthogonal training schemes such as Cayley transforms, making it unclear what unique benefits QR decomposition provides in the training context.

3.  The core contribution of integrating QR decomposition to enforce orthogonality in rotation-based pruning appears as a straightforward extension of existing rotation methods rather than a fundamentally novel technique. Without deeper theoretical insights or comparative analysis demonstrating why QR decomposition offers unique advantages over other manifold optimization schemes, this addition risks being perceived as incremental.

[1] "SpinQuant: LLM quantization with learned rotations", ICLR 2025

---

> ### Author Rebuttal · Authors · 2025-07-29
>
> Thank you for your detailed and thoughtful review of our paper. We appreciate your recognition of the novel perspective our framework, DenoiseRotator, brings to the field of neural network pruning for large language models. By focusing on importance redistribution rather than traditional weight selection, we aim to provide a significant departure from existing paradigms, supported by a theoretically well-motivated entropy-guided approach.
>
> We understand the concerns raised regarding the intuitive nature of pruning, which typically targets weights with low importance scores, and the need for a more thorough explanation of how entropy minimization contributes to improved pruning performance. We also acknowledge the need for a clearer comparative analysis of QR decomposition versus other orthogonal training methods, such as Cayley transforms, to elucidate the specific advantages offered by our approach.
>
> Your feedback is invaluable in helping us refine our explanations and strengthen the theoretical insights of our contributions. We are committed to addressing the weaknesses and questions you've highlighted to ensure the robustness and clarity of our work. Thank you again for your constructive comments and for guiding us toward enhancing the presentation and impact of our research.
>
> ### Weakness 1:
> Thank you for your thoughtful comments. The estimation of importance scores is crucial as they predict the change in output when parameters are removed, and they possess additive properties. Thus, the expected change in a layer's output due to pruning can be approximated by the sum of the importance scores of all pruned parameters. To minimize the change in output caused by pruning, it is essential to minimize the sum of the importance scores of the pruned parameters.
>
> In lines 182 to 195 and Appendix B of our paper, we provide a detailed explanation showing that the sum of importance scores for all parameters remains consistent before and after rotation. A lower-entropy weight distribution, compared to a uniform distribution, results in higher importance for top-ranked parameters and lower importance for bottom-ranked parameters (which are likely to be pruned). This leads to a smaller sum of importance scores for the pruned parameters. Consequently, the change in model output is minimized, making the model's output closer to the pre-pruning state and thus more amenable to pruning.
>
> We hope this explanation clarifies the rationale behind using entropy minimization to enhance pruning performance. Thank you for your feedback, which helps us improve the clarity and depth of our work.
>
> ### Weakness 2 & 3:
>
> Thank you for raising this important point regarding the comparative advantages of QR decomposition versus other orthogonal training schemes such as Cayley SGD. To address this, we conducted comparative experiments to evaluate the effectiveness of different optimizers:
>
> Configuration: LLaMA-3-8B, unstructured 50%, 2000 steps, evaluated by WikiText2 test set perplexity. Experiments are conducted on an A100 GPU, where the average time per step for the QR-based optimizer was 0.124 seconds, while for Cayley SGD it was 0.118 seconds.
>
> | Optimizer - Learning Rate \ Method |SparseGPT| Wanda.    |
> |------------------------------------|---------|-----------|
> | QR decomposition - 0.01                          | 7.597   | 7.816     |
> | Cayley SGD - 1                     | 7.727   | 9.695     |
> | Cayley SGD - 0.1                   | 8.097   | 9.827     |
> | Cayley SGD - 0.01                  | 8.832   | 9.834     |
>
> The results indicate that QR decomposition with a learning rate of 0.01 consistently outperforms Cayley SGD across different learning rates, in terms of perplexity.
>
> The superior performance of QR decomposition may be attributed to the fact that the actual optimization takes place in  $\mathbb{R}^{n \times n}$. In contrast, Cayley SGD operates within the Stiefel manifold, which is a subset of $\mathbb{R}^{n \times n}$. Therefore, QR decomposition allows for a broader exploration of the parameter space.
>
> We hope this comparative analysis clarifies the unique benefits of QR decomposition in the training context. Thank you for your feedback, which has helped us strengthen the clarity and depth of our research findings.

---

> > ### Author Response · Authors · 2025-08-04
> > **Looking forward for further feedback**
> >
> > Dear Reviewer 3pAp,
> >
> > Thank you once again for your thoughtful and encouraging comments on our submission!
> >
> > We have carefully addressed most of your concerns and are grateful for your valuable suggestions. We will incorporate all of these improvements in the revision.
> >
> > We are actively participating in the Author–Reviewer Discussion phase and would be happy to clarify or elaborate further on any aspect of our work.
> >
> > We look forward to your response!
> >
> > Best regards,
> >
> > Authors of Paper 9077

---

> > > ### Author Response · Authors · 2025-08-05
> > >
> > > Dear Reviewer 3pAp,
> > >
> > > We sincerely appreciate your time and effort in reviewing our manuscript and providing constructive feedback. Your insightful comments have been invaluable in helping us strengthen our work.
> > >
> > > In our rebuttal, we have conducted detailed comparative experiments to verify the superiority of QR decomposition-based orthogonal matrix optimization and provided a possible theoretical explanation. If you have any further questions or require additional clarification, please feel free to reach out.
> > >
> > > We would greatly appreciate it if you could consider increasing your score based on the additional insights provided.
> > >
> > > Best regards,
> > >
> > > Authors of Paper 9077

---

> > > > ### Comment · Reviewer_3pAp · 2025-08-07
> > > >
> > > > Thank you very much for your detailed response. I sincerely appreciate the authors' efforts in addressing my concerns. Your clarifications have successfully resolved my main concerns, and I found your explanations both thorough and convincing. Based on your response and the additional insights you provided, I am persuaded by your arguments. Therefore, I would like to raise my score.

---

### Official Review · Reviewer_BWCC · 2025-07-03

**Clarity:** 3
**Significance:** 2
**Originality:** 3
**Rating:** 4
**Confidence:** 5

**Summary:**

This paper proposes **DenoiseRotator**, a novel framework to improve the robustness of pruning in large language models (LLMs) by concentrating parameter importance *prior* to pruning. Unlike previous methods that only select which weights to prune, DenoiseRotator applies **learnable orthogonal transformations** to redistribute importance scores, guided by entropy minimization. This makes the model more amenable to pruning. The method is compatible with existing pruning algorithms (e.g., SparseGPT, Wanda, Magnitude) and is extensively evaluated on multiple LLMs (LLaMA3, Qwen2.5, Mistral) across different sparsity types (50% unstructured and 2:4 semi-structured). Experiments show consistent improvements in perplexity and zero-shot accuracy with minimal inference overhead.

**Questions:**

1. **Can this method be applied to structured pruning?** If yes, please provide results. If no, clarify why this is not feasible. This affects practical applicability on hardware with structured sparsity support.
2. **What is the optimal number of entropy reduction steps?** As shown in Table 3, over-training might hurt generalization (e.g., zero-shot accuracy drops after 2000 steps). Would an adaptive early stopping criterion help?

**Ethical Concerns:**

["NO or VERY MINOR ethics concerns only"]

**Final Justification:**

While the rebuttal addresses many of my concerns, some practical aspects—particularly around structured pruning and adaptive optimization—remain somewhat open in my view. I will therefore maintain my original score.

**Limitations:**

Yes, but it's not enough.

**Quality:**

3

**Strengths And Weaknesses:**

### Strengths
1. **Clear writing and explanation**: The paper is clearly written, and the methodology—including Figure 1 and the entropy-based formulation—is intuitive and easy to follow.
2. **Strong empirical results**: The method is extensively validated across different model scales, benchmarks, and pruning schemes. Visualization and ablations further support the effectiveness of the approach.
3. **Original idea**: The use of orthogonal rotations—originally popularized in quantization—for pruning is novel. Applying entropy-guided transformations to improve robustness is both theoretically grounded and practically effective.

---

### Weaknesses
1. **Limited practical efficiency in real deployment**: While effective under unstructured and semi-structured sparsity, the actual *inference speedup* under 2:4 patterns is limited. The method has not been shown to work under *structured pruning* (e.g., heads or channels), which is typically more impactful for latency reduction.
2. **Additional training cost**: DenoiseRotator requires a dedicated optimization phase for learning the orthogonal transformations. Although the cost is manageable, it adds extra complexity, potentially limiting usability in low-resource or deployment-constrained settings.

---

> ### Author Rebuttal · Authors · 2025-07-29
>
> Thank you for your thorough review and insightful feedback on our paper. We are grateful for your recognition of the clarity and originality of our work, DenoiseRotator, and the strong empirical results we have presented. Your appreciation of our approach, particularly the novel application of orthogonal rotations and entropy-guided transformations to enhance pruning robustness, is highly encouraging.
>
> We acknowledge the concerns regarding the limited practical efficiency in real deployment, particularly the inference speedup under 2:4 sparsity patterns and the applicability of our method to structured pruning. These are important areas that we are keen to address. Additionally, we understand the concern about the additional training cost associated with learning orthogonal transformations and its potential impact on usability in resource-constrained environments. Your questions about the feasibility of applying our method to structured pruning and the optimal number of entropy reduction steps are crucial for further enhancing the practical applicability of our approach. We will provide detailed responses and insights to clarify these aspects.
>
> Please find below our point-by-point response regarding your feedback:
>
> ### Weakness 1:
> Thank you for your insightful comments regarding the practical efficiency of our method in real deployment scenarios. We recognize that achieving inference speedup through model compression often involves trade-offs, where some model performance may be sacrificed. While structured pruning can more readily provide acceleration benefits, it typically results in more severe performance degradation compared to unstructured and semi-structured pruning. On the other hand, thanks to the specialized circuit design in NVIDIA GPUs, semi-structured sparsity can also effectively reduce inference time. We believe it is fairer to compare acceleration effects under similar levels of performance loss.
>
> The approach proposed in SliceGPT[1], which involves performing principal component analysis on model inputs to achieve structured pruning, serves as a valuable comparison point.
>
> Experiments comparing with SliceGPT:
> Configuration: calibration dataset consisting of 1024 samples of length 2048 from WikiText2, evaluating perplexity on the WikiText2 test set. SliceGPT's results are referenced from the paper.
>
> | Method                        | LLaMA-2-7B | LLaMA-2-13B | LLaMA-2-70B |
> |-------------------------------|------------|-------------|-------------|
> | Dense                         | 5.47       | 4.88        | 3.32        |
> | SparseGPT 2:4                 | 8.69       | 7.07        | 4.98        |
> | SparseGPT 2:4 + DenoiseRotator| 7.35       | 6.16        | 4.36        |
> | SliceGPT 10%                  | 5.89       | 5.21        | 3.69        |
> | SliceGPT 20%                  | 6.64       | 5.81        | 4.25        |
> | SliceGPT 25%                  | 7.24       | 6.30        | 4.60        |
> | SliceGPT 30%                  | 8.12       | 6.99        | 5.05        |
>
> The performance of the semi-structured sparse model pruned using the SparseGPT 2:4 + DenoiseRotator combination is comparable to that of the 25% structured pruning achieved by SliceGPT. However, the former requires less memory, only 60% of the original model, compared to 80% for the latter (SliceGPT adds two additional orthogonal matrices to each layer of the model). Furthermore, according to the Inference Time section in SliceGPT's main text, and Appendices A.6 and A.7, the latency of 2:4 semi-structured pruning is lower than that of 25% structured pruning.
>
> In summary, the semi-structured sparse model pruned using the combination of SparseGPT and DenoiseRotator achieves comparable performance to the structured sparse model pruned using SliceGPT 25%, while offering faster inference speed.
>
> ### Weakness 2:
> Thank you for highlighting the concern regarding the additional training cost associated with DenoiseRotator. We would like to clarify that the training resources required by DenoiseRotator are minimal, substantially less than those needed for actual deployment. Therefore, large language models capable of operating in resource-constrained or deployment-limited environments can also utilize DenoiseRotator for pruning within the same settings.
>
> Due to the independence of rotation matrices across layers, the training of each layer's rotation matrix can be conducted separately. The actual necessary memory consumption is calculated as 16 (comprising 4 for parameters, 4 for gradients, and 8 for optimizer states) multiplied by the hidden size squared, equivalent to the memory required for deploying one single layer. In our straightforward, uncompiled implementation, optimizing rotation matrices for LLaMA-3-8B requires only 10GB of memory, while LLaMA-3-70B requires only 30GB. Furthermore, the ablation study on training steps in Section 4.2, lines 253-273, demonstrates that even a small number of optimization steps can significantly enhance the performance of the pruned model.
>
> Additionally, model compression is a one-time process conducted before deployment, and the computational environment for compression does not need to match the deployment environment. Thus, the associated cost can be considered negligible.
>
>
> ### Question 1:
> Certainly, DenoiseRotator can be applied to structured pruning. However, it has been demonstrated in SliceGPT[1] that performing PCA on layer's input can yield the optimal rotation matrix suitable for structured pruning. Using DenoiseRotator to optimize the rotation matrix will ultimately converge to this optimal matrix, thus repeating previous research findings.
>
> For a comparison of the performance of semi-structured pruning using DenoiseRotator versus structured pruning, please refer to our response to Weakness 1.
>
> ### Question 2:
>
> You are absolutely right; employing an adaptive early stopping criterion can be an effective way to determine the appropriate number of optimization steps. However, due to the lack of interpretative analysis between model performance and intermediate hidden states, we are currently unable to provide a theoretical framework and must rely on experimental methods to determine this.
>
> To address this, we conducted a more rigorous grid search for hyperparameters (step, learning rate) on LLaMA-3-8B. Our findings revealed that zero-shot task accuracy does not exhibit a strict linear relationship with perplexity; rather, it appears to fluctuate randomly.
>
> In our experiments, the left column represents perplexity on the WikiText2 test set, while the right column shows average zero-shot accuracy. The experimental setup involves SparseGPT with 50% unstructured pruning, using a calibration dataset consisting of 128 samples of length 2048 from C4. All experiments were performed on an A100 80G GPU.
>
> The table below presents the results across various learning rates and training steps:
>
> | Steps\Learning Rate | 0.1         | 0.01        | 0.001       | 0.0001     |
> |---------------------|-------------|-------------|-------------|------------|
> | 100                 | 7.75\69.52  | 7.70\70.54  | 7.96\69.95  | 8.52\67.83 |
> | 200                 | 7.70\70.31  | 7.64\68.93  | 7.82\70.04  | 8.30\68.59 |
> | 400                 | 7.63\69.84  | 7.62\70.12  | 7.66\70.31  | 8.16\69.37 |
> | 800                 | 7.61\70.25  | 7.61\70.21  | 7.72\69.55  | 7.94\70.77 |
> | 2000                | 7.64\69.82  | 7.60\69.58  | 7.67\70.36  | 7.85\70.11 |
> | 4000                | 7.60\70.46  | 7.60\69.59  | 7.63\70.37  | 7.78\69.59 |
>
> These results indicate that the zero-shot task accuracy fluctuates around 70% with a variation of approximately 1%, not showing a clear linear relationship with perplexity.
>
> Summary: The optimal number of hyperparameters and the calibration dataset should be selected based on the model's intended application empirically. For example, a model designed to assist with coding should choose the optimal entropy reduction steps and calibration dataset to maximize coding capabilities. A model intended for chat should aim to maximize user experience. The required optimization steps vary across specific tasks and evaluation criteria; some may be fewer than 2000, while others may exceed 2000. Conducting a grid search for optimization steps is the simplest and most effective approach.
>
>
> [1]"SLICEGPT: COMPRESS LARGE LANGUAGE MODELS BY DELETING ROWS AND COLUMNS", ICLR 2024

---

> > ### Author Response · Authors · 2025-08-04
> > **Looking forward for further feedback**
> >
> > Dear Reviewer BWCC,
> >
> > Thank you once again for your thoughtful and encouraging comments on our submission!
> >
> > We have carefully addressed most of your concerns and are grateful for your valuable suggestions. We will incorporate all of these improvements in the revision.
> >
> > We are actively participating in the Author–Reviewer Discussion phase and would be happy to clarify or elaborate further on any aspect of our work.
> >
> > We look forward to your response!
> >
> > Best regards,
> >
> > Authors of Paper 9077

---

> > ### Comment · Reviewer_BWCC · 2025-08-05
> > **Official Comment by Reviewer BWCC**
> >
> > Thank you for the detailed and thoughtful response. I appreciate the additional experiments and clarifications. While the rebuttal addresses many of my concerns, some practical aspects—particularly around structured pruning and adaptive optimization—remain somewhat open in my view. I will therefore maintain my original score.

---

### Official Review · Reviewer_5XD8 · 2025-07-05

**Clarity:** 3
**Significance:** 3
**Originality:** 4
**Rating:** 4
**Confidence:** 3

**Summary:**

This paper introduces DenoiseRotator, a novel framework designed to enhance the robustness of LLM pruning. The core thesis is to shift the paradigm from simply selecting which weights to prune to first preparing the model to be more prunable. This is accomplished by using learnable orthogonal transformations to actively concentrate parameter importance into a smaller subset of weights. The optimization process is guided by minimizing the information entropy of normalized importance scores. DenoiseRotator is presented as a plug-and-play module that is compatible with existing pruning techniques like Wanda and SparseGPT. Extensive experiments on models like LLaMA3 and Qwen2.5 demonstrate significant improvements in perplexity and zero-shot accuracy under both unstructured and 2:4 semi-structured sparsity.

**Questions:**

1. The results in Table 3 hint at a potential trade-off between perplexity and zero-shot accuracy as training progresses. Could you provide more insight into this behavior? Does it suggest a risk of overfitting to the calibration set, and is there an optimal stopping point for the rotation training?
2. Could the authors elucidate whether solving eq(6) is also NP-H and how does this corresponds to eq(4)? Does other variance concentration techniques works, such as PCA on the centralized covariance of S?
3. How does the importance of weight indices corresponds to the transformed scores?

I am willing to adjust my ratings after seeing the authors' responses.

**Ethical Concerns:**

["NO or VERY MINOR ethics concerns only"]

**Final Justification:**

I think all my concerns have been addressed and I would like to maintain my current recommendations.

**Limitations:**

Yes

**Quality:**

4

**Strengths And Weaknesses:**

Strengths:

1.Novelty of the Core Idea: The idea of importance concentration as a preprocessing step is a significant and insightful departure from traditional pruning pipelines, offering a new and promising direction for model compression research.

2.High Practicality and Compatibility: By decoupling the importance reshaping from the pruning algorithm itself, DenoiseRotator is designed as a versatile, plug-and-play module. This significantly lowers the barrier to adoption for researchers and practitioners already using established pruning workflows.

3.Strong and Comprehensive Empirical Results: The authors provide compelling evidence across multiple model families (LLaMA3, Qwen2.5, Mistral) and sizes. The consistent and significant performance gains in both perplexity and zero-shot tasks, especially under the hardware-relevant 2:4 sparsity, strongly validate the method's effectiveness.

Weaknesses:
1.Indirect Mechanism for Structured Sparsity: The paper's justification for its strong performance on 2:4 sparsity is presented as a hypothesis—that the rotation acts like a "random permutation" improving alignment. This suggests the entropy-minimization objective is not explicitly tailored for structured pruning, and the precise mechanism behind its success in this crucial scenario is not fully understood.

2.Potential Overstatement of "Model-Agnostic" Claim: While tested on several decoder-only models, the paper's claim of being "model-agnostic" may not fully extend to other architectures like encoder-decoder or Mixture-of-Experts (MoE) models without further investigation.

3.Limited Analysis of Hyperparameter Sensitivity: The paper reports results using a fixed set of hyperparameters for training the rotations (e.g., 2000 steps). A more thorough analysis of how sensitive the final performance is to these choices would strengthen the paper's conclusions and improve its reproducibility.

---

> ### Author Rebuttal · Authors · 2025-07-29
>
> Thank you for taking the time to review our paper and for providing such detailed and constructive feedback. We greatly appreciate your recognition of the novelty and practical implications of our work, DenoiseRotator, as well as the comprehensive empirical results we have presented. Your insights into the strengths of our approach, particularly the innovative concept of importance concentration and its compatibility with existing pruning techniques, are invaluable to us.
>
> Please find below our point-by-point response regarding your feedback:
>
> ### Weakness 1:
>
> The notable performance in 2:4 semi-structured pruning is not solely due to the rotation acting as a "random permutation matrix." Instead, the "random permutation" is an integral part of the rotation matrix-based importance concentration mechanism, both of which contribute to the enhancement of performance.
>
> 2:4 semi-structured pruning represents a constrained version of unstructured pruning. In the context of importance score-based pruning, the primary difference between the two lies in the selection of comparison groups. Specifically, 2:4 semi-structured pruning eliminates the two least important parameters from every four consecutive parameters, whereas unstructured pruning chooses parameters to prune from a larger group (SparseGPT sets its pruning group to 256 columns, Wanda to 1 column).
>
> The importance concentration mechanism is designed to intensify the focus of importance within the normalization group (1 column or 1 row). For 50% unstructured pruning, half of the low-importance parameters need to be removed from the entire comparison group (containing multiple normalization groups). In contrast, for 2:4 semi-structured pruning, the normalization group is divided into multiple blocks of four consecutive parameters, from which half of the low-importance parameters are removed. If each block retains two parameters from the 50% unstructured pruning and removes two, the 2:4 semi-structured pruning will mirror the effect of 50% unstructured pruning, making the importance concentration mechanism equally effective. However, in practice, there may be blocks with three or four important parameters, which could lead to inferior performance in 2:4 pruning compared to 50% unstructured pruning.
>
> To ensure that parameters retained in 50% unstructured pruning are more evenly distributed across the blocks of 2:4 semi-structured pruning, a permutation matrix can be utilized to swap parameters among these blocks. According to probability theory, a randomly generated permutation matrix can effectively achieve this task. Since permutation matrices are orthogonal, multiplying them with another orthogonal matrix results in an orthogonal matrix. Additionally, permutation does not affect entropy, allowing for numerous equivalent rotation matrices, with gradient descent randomly optimizing to one of them.
>
> In summary, the synergy between the importance concentration mechanism and random permutation collectively enhances the performance observed in 2:4 semi-structured pruning in this paper.
>
> ### Weakness 2:
>
> Thank you very much for pointing out the potential overstatement regarding the "model-agnostic" claim in this paper. We have indeed tested on several decoder-only models, but we recognize that this claim may not yet fully extend to other architectures such as encoder-decoder or Mixture-of-Experts (MoE) models. If this paper is accepted, we will promptly revise the text to ensure alignment with the experimental results and scope presented in the paper. Your feedback is greatly appreciated and will help us enhance the rigor and accuracy of our research.
>
> ### Weakness 3:
>
> Thank you very much for your suggestions regarding the analysis of hyperparameter sensitivity. We completely agree with your viewpoint that a thorough analysis of how hyperparameter choices affect final performance will help strengthen the conclusions of the paper and improve its reproducibility. The current paper indeed uses a fixed set of hyperparameters for experiments, such as 2000 steps in rotation training. To enhance the rigor of our research, we conducted a more comprehensive analysis on LLaMA-3-8B to assess the impact of different hyperparameter choices on the final results.
>
> In our experiments, the left column represents perplexity on the WikiText2 test set, while the right column shows average zero-shot accuracy. The experimental setup involves SparseGPT with 50% unstructured pruning, using a calibration dataset consisting of 128 samples of length 2048 from C4. All experiments were performed on an A100 80G GPU.
>
> The table below presents the results across various learning rates and training steps:
>
> | Steps\Learning Rate | 0.1         | 0.01        | 0.001       | 0.0001     |
> |---------------------|-------------|-------------|-------------|------------|
> | 100                 | 7.75\69.52  | 7.70\70.54  | 7.96\69.95  | 8.52\67.83 |
> | 200                 | 7.70\70.31  | 7.64\68.93  | 7.82\70.04  | 8.30\68.59 |
> | 400                 | 7.63\69.84  | 7.62\70.12  | 7.66\70.31  | 8.16\69.37 |
> | 800                 | 7.61\70.25  | 7.61\70.21  | 7.72\69.55  | 7.94\70.77 |
> | 2000                | 7.64\69.82  | 7.60\69.58  | 7.67\70.36  | 7.85\70.11 |
> | 4000                | 7.60\70.46  | 7.60\69.59  | 7.63\70.37  | 7.78\69.59 |
>
> These results indicate:
>
> 1. When the learning rate is less than or equal to 0.001, the efficiency is low due to the small learning rate.
> 2. When the learning rate is 0.01, the training converges around 2000 steps, and excessive steps increase training costs.
> 3. Compared to the baseline without the importance concentration mechanism, all configurations show significant improvement as training progresses.
> 4. The zero-shot task accuracy fluctuates around 70% with a variation of approximately 1%, not showing a clear linear relationship with perplexity.
>
> ### Question 1:
>
> Certainly, there appears to be a trade-off between perplexity and zero-shot accuracy as training progresses. The training of the rotation matrix can be viewed as a form of continuous pretraining on the calibration dataset. The observed trade-off between different tasks in the later stages of training aligns with observations in these domains, and the risk of overfitting to the calibration set can be referenced from studies in these fields. Unfortunately, due to the lack of open-source pretraining datasets for large models and the absence of a clear theoretical analysis on neural network interpretability, we cannot provide a theoretical framework to determine the optimal stopping point.
>
> Moreover, detailed experimental results analyzing hyperparameter combinations in Response to Weakness 3 show that zero-shot task accuracy exhibits slight fluctuations and does not have a strict linear relationship with perplexity. Therefore, the most effective approach is to use grid search to determine the training steps that optimize model performance in practical application scenarios.
>
> ### Question 2:
>
> Thank you for your insightful questions. Addressing whether solving equation (6) is NP-hard requires considering different scenarios. In Scenario 1, if the transformation $T_s$ for importance scores is continuously differentiable, then all components in equation (6) are continuously differentiable, allowing optimization via gradient descent according to the chain rule. In Scenario 2, if $T_s$ is not continuously differentiable, gradient descent cannot be used, and whether it is NP-hard depends on the number of possible $T_s$. DenoiseRotator uses continuously differentiable orthogonal matrix transformations for  $T_s$, falling under Scenario 1, thus it is not NP-hard.
>
> Equation (6) serves as a continuously differentiable substitute for equation (4), and optimizing equation (6) can optimize equation (4) empirically. For details, please refer to lines 127-135.
>
> Regarding the effectiveness of other variance concentration techniques, such as PCA on the centralized covariance of S, this is an excellent question. PCA is a well-known variance concentration technique primarily used for dimensionality reduction of high-dimensional data, which is assumed to be sampled from a high-dimensional random variable. However, parameters are a set of fixed constants, and the formula for parameter importance involves the parameters themselves and the expected value of corresponding inputs, lacking the essence of random sampling. Therefore, PCA should target the hidden dimension of layer's inputs that can be viewed as samples from high-dimensional random variables. Columns of the weight matrix corresponding to the inputs' pruned hidden dimensions are removed. Thus, PCA inherently implies structured constraints, requiring complete removal of entire columns. In fact, SliceGPT[1] employs this approach using PCA for structured pruning. The importance concentration mechanism proposed in this paper, based on entropy minimization, does not have structured constraints, offering greater flexibility and a higher performance ceiling than PCA. Experimental results also demonstrate that our proposed method achieves superior performance.
>
> ### Question 3:
> Thank you for your inquiry regarding the relationship between the importance of weight indices and transformed scores. The importance of weight indices is assessed based on the contribution of parameters to the model's predictive capabilities. Transformed scores, however, represent the importance of weight indices after they have been modified through rotation or other transformation techniques to facilitate more robust pruning. In this paper, transformations are implemented using rotation matrices, and the transformed importance can be calculated from the original weights and the current rotation matrix using the formula provided in Appendix A.
>
> [1]"SliceGPT: Compress Large Language Models by Deleting Rows and Columns", ICLR 2024

---

> > ### Author Response · Authors · 2025-08-04
> > **Looking forward for further feedback**
> >
> > Dear Reviewer 5XD8,
> >
> > Thank you once again for your thoughtful and encouraging comments on our submission!
> >
> > We have carefully addressed most of your concerns and are grateful for your valuable suggestions. We will incorporate all of these improvements in the revision.
> >
> > We are actively participating in the Author–Reviewer Discussion phase and would be happy to clarify or elaborate further on any aspect of our work.
> >
> > We look forward to your response!
> >
> > Best regards,
> >
> > Authors of Paper 9077

---

> > ### Comment · Reviewer_5XD8 · 2025-08-04
> >
> > Thank the authors for their responses to my concerns/questions. All of my concerns are well addressed. I would like to raise my score to accept.

---

> > > ### Author Response · Authors · 2025-08-05
> > >
> > > We sincerely appreciate your time and effort in reviewing our manuscript and providing constructive feedback. Thank you for recognizing our responses in the rebuttal and for updating your evaluation accordingly. We are truly grateful for your insightful comments, which have helped strengthen our work.
> > >
> > > Your expertise and thoughtful review are invaluable to us and the peer-review process. We deeply appreciate your consideration and support.
> > >
> > > Kindly remember to update your score in the system and confirm the Mandatory Acknowledgement.
> > >
> > > Best regards,
> > >
> > > Authors of Paper 9077

---

### Note · Authors · 2025-08-12

Dear Reviewers, Area Chairs, Senior Area Chairs, and Program Chairs,

We sincerely thank all reviewers for their positive feedback and constructive comments. Reviewers have acknowledged the originality of our approach, its high practicality and compatibility, and the significant performance gains validated through extensive experiments.

**Key Highlights:**

- **Novelty**: Importance concentration represents a promising direction for model compression (5XD8). Entropy-minimized importance shaping is new in pruning literature (PoQx).
- **Practical Methods**: Compatible with existing pruning algorithms and allows easy integration (BWCC, 3pAp).
- **Quality**: Solid implementation with QR re-parameterization and strong empirical results (PoQx, BWCC).
- **Comprehensive Experiments**: Compelling evidence across multiple model families and sparsity patterns (5XD8, PoQx).
- **Performance**: Consistent improvements in perplexity and zero-shot accuracy (BWCC).

During the past reviews, we have worked diligently to improve experiments, clarifications, and discussions to address the concerns raised by all reviewers. Specifically, we have:

(1) Provided detailed comparative analysis and clarity on theoretical motivations, confirming the advantages of QR decomposition over alternative methods.

(2) Enhanced discussions on recovery fine-tuning strategy compatibility, ensuring practical utility without impacting training stability.

(3) Addressed questions regarding structured sparsity with empirical evidence and theoretical explorations.

(4) Conducted more detailed hyper-parameter analysis experiments.

(5) Explored options to reduce overhead, including the use of block-diagonal matrices.

We will carefully integrate all these improvements into a cohesive revision that enhances the overall quality of our work. We hope our response addresses the remaining concerns, and we thank the reviewers again for their helpful comments. We are glad to discuss further comments and suggestions.

Best regards,

Authors of Paper 9077

---

### Decision · Program_Chairs · 2025-09-17

**Decision:**

Accept (poster)

**Comment:**

This paper proposes a learnable rotation on the weights using entropy minimization, allowing important parameters to concentrate into small subsets of weights, which improves pruning.

Strengths
- Interesting idea of rotating the weight matrices.
- Can be integrated with other pruning methods.
- Solid empirical results.

Weaknesses
- Potentially limited to semi-structured pruning.
- Additional training cost for learning the rotation.
- Possible overlap with the idea of RotPruner.

During the rebuttal, the authors presented additional experiments, including:
- Ablation studies on the hyperparameters.
- Demonstration of gains from semi-structured pruning.
- Clarification that the additional training introduces minimal overhead.
- Comparison with other orthogonal training schemes.
- Use of other structured matrices for rotation.
- Comparison with RotPruner.
- Further experiments on fine-tuning.

I agree with the reviewers that the paper should be accepted.